# Resolving mixed mechanisms of protein subdiffusion at the T cell plasma membrane

Yonatan Golan[1] & Eilon Sherman[1]

The plasma membrane is a complex medium where transmembrane proteins diffuse and interact to facilitate cell function. Membrane protein mobility is affected by multiple mechanisms, including crowding, trapping, medium elasticity and structure, thus limiting our ability to distinguish them in intact cells. Here we characterize the mobility and organization of a short transmembrane protein at the plasma membrane of live T cells, using single particle tracking and photoactivated-localization microscopy. Protein mobility is highly heterogeneous, subdiffusive and ergodic-like. Using mobility characteristics, we segment individual trajectories into subpopulations with distinct Gaussian step-size distributions. Particles of low-to-medium mobility consist of clusters, diffusing in a viscoelastic and fractal-like medium and are enriched at the centre of the cell footprint. Particles of high mobility undergo weak confinement and are more evenly distributed. This study presents a methodological approach to resolve simultaneous mixed subdiffusion mechanisms acting on polydispersed samples and complex media such as cell membranes.

[1] Racah Institute of Physics, The Hebrew University, Jerusalem 91904, Israel. Correspondence and requests for materials should be addressed to E.S. (email: sherman@phys.huji.ac.il).

The plasma membrane (PM) of cells is a diverse, multi-component complex medium through which the cell interacts with its surroundings. Proteins at the PM diffuse and interact to facilitate a wide range of cellular functions, including sensing and signalling[1]. Specifically, T cells probe the surface of antigen-presenting cells (APCs) for cognate antigens to trigger an adaptive immune response. Antigen recognition is achieved by highly specific T-cell antigen receptors (TCRs) and leads to the rapid development of a complex interface between the cells—the immune synapse. T-cell activation results in dramatic macroscopic rearrangement of protein distribution at the immune synapse[2,3]. However, much remains to be learned about the microscopic properties of the PM and protein mobility within[4].

According to the Nicholson–Singer model, the PM can be regarded as a complex fluid, in which transmembrane proteins diffuse laterally[5]. The random motion of particles in a purely viscous and homogeneous fluid is known as Brownian motion and is characterized in two dimensions by:

$$\langle \Delta r^2(t) \rangle = \int_{-\infty}^{\infty} r^2(t) P(r,t) \mathrm{d}^2 r = 4Dt \qquad (1)$$

where the left-hand side is the mean squared displacement (MSD) of the particle from its origin, $P(r,t)$ is the probability distribution function (PDF; or propagator) of the diffusion process, $D$ is the diffusion constant and $t$ is time. The MSD is typically measured in two ways. The first, as an average across an ensemble of particles (eMSD):

$$\langle \Delta r^2(t) \rangle_{\text{ens}} = \frac{1}{N} \sum_{i=1}^{N} (\vec{r}_i(t) - \vec{r}_i(0))^2 \qquad (2)$$

where $\vec{r}_i(t)$ is the location of particle $i$ at time $t$ and $N$ is the number of particles in the ensemble. The second method, as a function of gap time for a single particle (tMSD):

$$\langle \Delta r^2(\tau \cdot \Delta t) \rangle_{\text{time}} = \frac{1}{T-\tau} \sum_{i=1}^{T-\tau} (\vec{r}_{i+\tau} - \vec{r}_i)^2 \qquad (3)$$

where $T$ is the number of frames in the trajectory, $\tau$ is the time gap measured in frames and $\Delta t$ is the measurement time step. One can further take the mean of the tMSD functions of multiple trajectories to obtain an average tMSD of the ensemble:

$$\left\langle \langle \Delta r^2(\tau) \rangle_{\text{time}} \right\rangle_{\text{ens}} = \frac{1}{N} \sum_{i=1}^{N} \langle \Delta r^2(\tau) \rangle_{\text{time},i} \qquad (4)$$

The ensemble and mean time averages converge to the same value for large $N$ and $t$ for an ergodic system.

Complex media may lead to sublinearity of the MSD as a function of time[6–9], that is,

$$\langle \Delta r^2(t) \rangle = K_\alpha t^\alpha, \qquad (5)$$

where $0 \le \alpha < 1$. Such a motion is said to be subdiffusive. $K_\alpha$ is the generalized diffusion coefficient with dimensions of $[\mathrm{m}^2 \mathrm{s}^{-\alpha}]$. Note should be taken that the mean of the tMSD functions in equation (4) is an arithmetic mean. In contrast, a geometric mean provides an accurate estimator for the mean value of the power $\alpha$ (ref. 10):

$$\left\langle \langle \Delta r^2(\tau) \rangle_{\text{time}} \right\rangle_{\text{ens}} = \left( \prod_{i=1}^{N} \langle \Delta r^2(\tau) \rangle_{\text{time},i} \right)^{1/N}$$

$$= \exp\left( \sum_{i=1}^{N} \log\left( \langle \Delta r^2(\tau) \rangle_{\text{time},i} \right) \right) \qquad (6)$$

The geometric mean is mathematically smaller or equal to the arithmetic mean. Because of the differences in averaging, the resultant $K_\alpha$ values for the time averaged measurements in equation (6) are biased to be smaller than the $K_\alpha$ values of the ensemble averages in equation (2).

Several mechanisms may give rise to subdiffusive motion. These are often described by related mathematical models, including[11–13]: (a) Diffusion of tracer particles in a viscoelastic medium statistically results in anti-persistent motion and can be described using the fractional Brownian motion (fBM) model[14]; (b) Tracer particles may experience trapping by specific interactions with other particles or objects in the medium. The particles may exhibit trapping events with a heavy-tailed waiting time distribution. Such motion can be described using the continuous time random walk (CTRW) model[15]; (c) Tracer particles diffusing in obstructed or labyrinth-like environments demonstrate movement in a fractal-like space with a dimension $d_f$ smaller than the real space dimension. Such movement is modelled by a random walk on a fractal (RWF)[16]; (d) Tracers diffusing in a confined environment due to non-permeable physical boundaries demonstrate normal diffusion within the boundaries at short timescales, appear to be subdiffusive in intermediate timescales, and will saturate to a flat MSD at long timescales. In the case of permeable boundaries, the MSD will regain normal diffusion at long timescales[17].

A set of tests has been proposed to determine the dominating mechanism underlying an observed subdiffusive process[11,18]. Naturally, multiple mechanisms of subdiffusion may act simultaneously, which may dramatically complicate the analysis. For instance, cases of mixed mechanisms have been demonstrated and modelled by a combination of CTRW and RWF processes[19,20]. Moreover, particle mobility may be also complicated by static heterogeneity (for example, particle polydispersity) or spatial inhomogeneity of the medium. Effective or unified models have been developed to address heterogeneity in single particle tracking (SPT) results[21–26]. However, cases of mixed underlying mechanisms of subdiffusion or of static heterogeneity remain poorly understood as it is often unclear how to distinguish between the underlying mechanisms.

Here we characterize the mobility and organization of a short transmembrane protein at the PM of live T cells, using SPT[27] and photoactivated-localization microscopy (PALM). This transmembrane segment shows highly heterogeneous subdiffusive mobility consistent with ergodic underlying mechanisms. We use a segmentation approach combined with multiple statistical tests[11,28–30] to discriminate and characterize subdiffusive behaviour, including fractal-like and weak confinement. This method is able to resolve simultaneous mixed subdiffusion mechanisms acting on cell membranes.

## Results

**SPT of a short transmembrane protein.** The tracers we use are part of the human immunodeficiency virus envelope glycoprotein gp41, whose assembly at the PM is critical for viral budding[31]. Our construct consists of the transmembrane domain of gp41 with a mutation that inhibits its binding to cholesterol[32] and thus, to cholesterol-enriched domains at the PM. The gp41 constructs are tagged with a fluorescent protein (PAGFP), which is specifically labelled with a primary antibody stained with the fluorescent dye Alexa Fluor 594 (AF594). Using total internal reflection (TIRF) microscopy, we visualize and localize protein-containing particles with a resolution of $\sim 25$ nm at a frame rate of 85 fps (Supplementary Fig. 1a). Importantly, our constructs could be imaged for relatively long durations (Supplementary Fig. 1b) at a high signal-to-background ratio using either AF594

fluorescence for SPT or PAGFP photoactivated emission in PALM mode[33] (see 'Methods' section). Figure 1a shows a representative frame of an imaged live T cell as it adheres to a functional TCR-stimulating coverslip. The coverslip is coated with an αCD3ε antibody that crosslinks the TCRs and leads to robust cell activation and spreading. The bright spots are particles highlighted by AF594 at the PM of the cell. The inset shows the bright field image of the cell. Imaging of cells over 59 s (5,000 frames) yields movies in which particles move across the cell footprint. Next, we use a tracking algorithm (modified from refs 34,35) to track single particles for durations of up to 5,000 frames before fluorescence photobleaching of the sample. Trajectories shorter than 50 frames are discarded in the analysis to increase the robustness of the data. Figure 1b,c shows a representative cell with trajectories coloured by the values of their individual subdiffusive power α. A wide range of particle mobility is detected. Figure 1d,e shows a PALM image of a representative cell.

**Protein mobility is heterogeneous and ergodic-like**. We next aim to study the mobility of individual particles and their underlying subdiffusion mechanisms. However, we notice a large fraction of immobile trajectories that may originate from non-specific antibody staining, background or completely immobile particles. These trajectories may complicate our understanding of the system due to their dominating abundance relative to the rest of the mobile trajectories. Thus, we start our analysis by removing completely immobile trajectories. Previous studies achieved this by placing a threshold on the diffusion coefficient[22], or on the radius of gyration $R_g$ of the trajectory[21], which is calculated as:

$$R_g^2 = \frac{1}{N}\sum_{i=1}^{N}(\vec{r}_i - \langle\vec{r}\rangle)^2 \qquad (7)$$

where $\vec{r}_i$ is the coordinate at time step $i$ out of $N$ total time steps. We consider the case of an immobile particle in which the course of the apparent trajectory is governed solely by the localization error of the system[36,37]. In two dimensions, a particle located at the origin would result in localizations distributed normally with mean 0 and variance $\sigma^2$. In such a case, $R_g^2$ and the mean step size $\langle|\Delta r|\rangle$ are $2\sigma^2$ and $\sigma\sqrt{\pi}$, respectively (see 'Methods' section). Thus, the ratio $R_g/\langle|\Delta r|\rangle = \sqrt{2/\pi}$ is a constant independent of the localization error. Therefore, we use here both $R_g$ and $\langle|\Delta r|\rangle$ of the trajectory to robustly remove immobile particles. We validate our approach by comparing two controls (immobile gold beads and gp41 proteins in a fixed sample) and compare their results to data acquired in live cells (Fig. 2a). The normalized ratio $\sqrt{\pi/2}R_g/\langle|\Delta r|\rangle$ of the controls serves to set a threshold for robustly excluding immobile particles (66% of all detected particles in the live cells; see 'Methods' section).

We now turn to the characterization of the mobile particles through their time and ensemble MSDs (Fig. 2b). From the MSDs we calculate α, which is the slope of the curve in logarithmic scale, and $K_\alpha$, which is its value at $t=1$. We find that on average, the mobility of particles is subdiffusive ($\alpha_t = 0.72 \pm 0.04$; $\alpha_e = 0.75 \pm 0.01$). As expected from the differences between the arithmetic and geometric averaging, we obtain $K_{\alpha,t} = 1.09(1) \times 10^{-13}$ m$^2$ s$^{-\alpha}$, which is smaller than $K_{\alpha,e} = 3.4(1) \times 10^{-13}$ m$^2$ s$^{-\alpha}$. To account for this difference, we simulate MSDs with comparable characteristics to the experimental data and find similar differences in $K_\alpha$ values between the arithmetic and geometric means (Supplementary Fig. 2). The small difference in the experimental α values of the time and ensemble averages (being significantly different from 1.0) are consistent with ergodic underlying mechanisms of subdiffusion. We suggest later in the discussion that this small difference is due to weak confinement of the particles. Furthermore, our data shows high

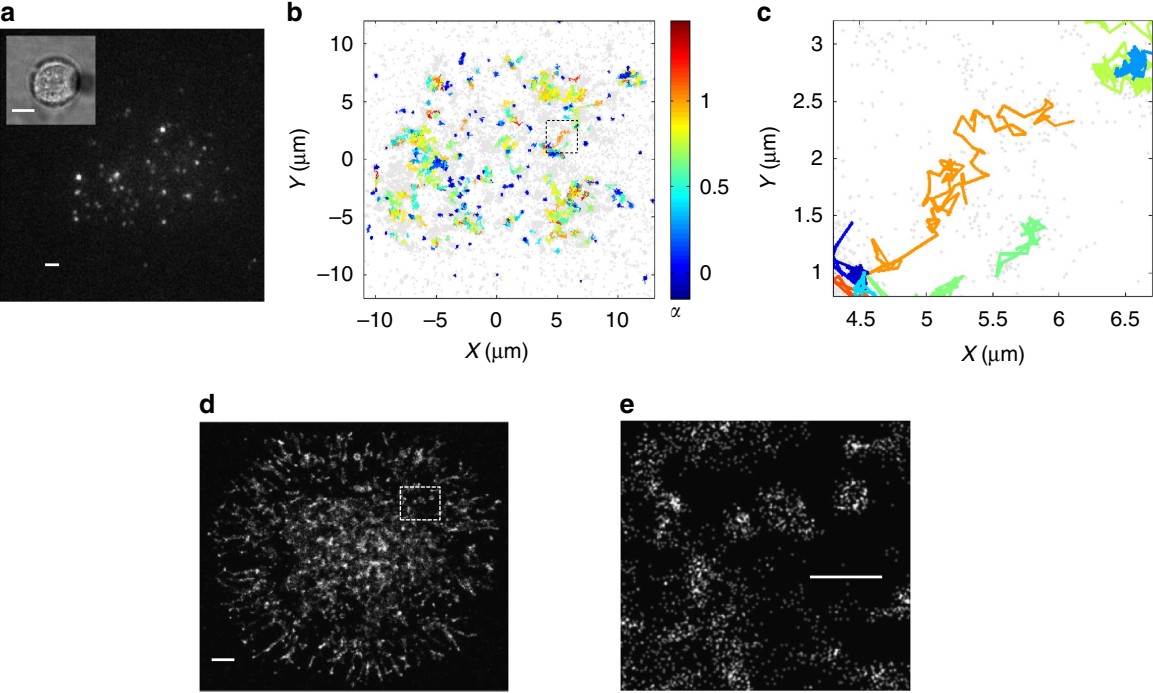

**Figure 1 | Single particle imaging and tracking in live T cells.** (**a**) A representative video frame showing gp41 constructs tagged with AF594 at the PM of a live, activated Jurkat T cell as it spreads on an αCD3ε-coated coverslip that stimulates the TCR. Scale bar, 2 μm. Inset: a bright field image of the cell at × 40 magnification. Scale bar, 10 μm. (**b**) A representative spatial map of gp41 trajectories on the cell membrane. Trajectories longer than 50 frames (∼0.6 s) are shown and coloured by their individual subdiffusive power α. Particles not assigned to long enough trajectories are marked in grey and highlight the cell footprint. (**c**) Enlarged area of spatial map in **b**. (**d**) PALM image of PAGFP-tagged gp41 constructs in a representative fixed T cell. Scale bar, 2 μm. (**e**) Enlarged area of PALM image marked in **d**. Scale bar, 1 μm.

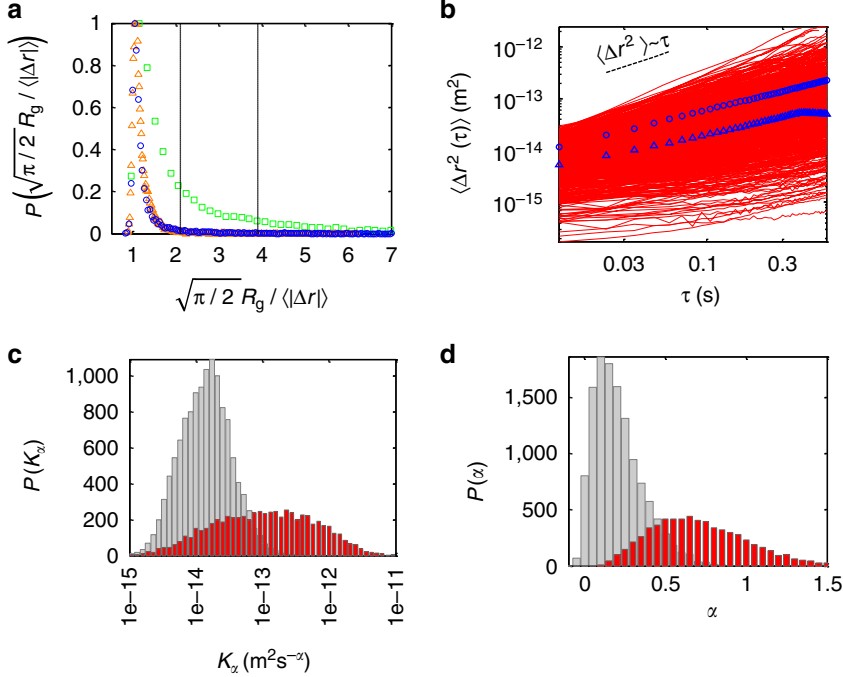

**Figure 2 | High heterogeneity of subdiffusive motion at the ensemble level.** (**a**) Distribution of the radius of gyration normalized by the mean step size for gp41 proteins marked with AF594 in live cells (green squares), gp41 proteins in fixed cells (blue circles) and fixed 40 nm gold beads (orange triangles). The two vertical lines indicate the values up to which 95 and 99% of the particles in the fixed cells reside—that is, the threshold values for 5 and 1% particles falsely considered as mobile (false positives). (**b**) A log–log plot of the MSD with respect to time for all mobile trajectories. Shown are tMSDs for single trajectories (red lines) (only 10% of data is shown for clarity), eMSD (blue circles) and the average of the individual tMSDs (blue triangles). (**c,d**) Distributions of **c** $K_\alpha$ and **d** $\alpha$ values for single trajectories. Both mobile and immobile trajectories (red and grey bars, respectively) are shown.

heterogeneity of $\alpha$ values from 0 to $\sim1.5$, and of $K_\alpha$ values, which span over $\sim4$ orders of magnitude (Fig. 2c,d).

**Resolving heterogeneity by trajectory segmentation**. Closer examination of single particle trajectories reveals heterogeneity on the individual trajectory level (dynamic heterogeneity; Fig. 3) and on the ensemble level (static heterogeneity) (Fig. 2c,d). Figure 3a shows representative trajectories that display dynamic alterations in the diffusion state of the tracer particle (highlighted by arrows). Previous studies have shown or suggested such dynamic heterogeneity in the motion of transmembrane proteins in which particles can alternate their mobility state during their motion[21,38,39].

To better understand the underlying mechanisms of sub-diffusion and to resolve possible dynamic heterogeneity, we perform segmentation of individual trajectories into separate parts that correspond to different mobility states $\{s_i\}$. For convenience, we order the states in this set in an ascending order according to their mobility characteristics (that is, state $i+1$ is more mobile than state $i$). Note that this ordering assumes no restriction on the possible transition of a particle from one state to any other state.

Multiple approaches have been suggested for segmentation of particle trajectories, often by the identification of transitions between states[26,40]. Trajectory segments with similar characteristics are then assigned to the same state. These segments can be grouped together into subpopulations for further study of their underlying mechanism. Since there are generally no 'closed-form' guidelines for performing optimal segmentation, we use two complementary approaches that highlight different aspects of the data, as explained below.

The first method is a variational Bayesian treatment of a hidden Markov model (vbSPT)[25]. Briefly, this method uses a

maximum-evidence criterion to determine the underlying parameters and the number of diffusive states from the observed data. This method can robustly resolve multiple underlying diffusion states from a large number of short trajectories. However, it assumes only pure Brownian motion for each state.

As a second segmentation approach, we adapt concepts used in 'First passage time' or 'escape radius' statistics[19,41,42], and introduce a modified method that we term 'consecutive escape radii' (CER). Through this method, we measure the particle's ability to travel a given distance in a given time. For each time point in the trajectory, we test if the particle exceeds a distance $R_{th}^{s_i,s_{i+1}}$ during its next $n_{th}^{s_i,s_{i+1}}$ time steps. The writing $s_i,s_{i+1}$ above a variable of interest denotes the threshold value for discriminating between states $i$ and $i+1$ in the ordered set of states $\{s_i\}$. Figure 3b demonstrates this segmentation method for three states $\{s_1,s_2,s_3\}$ and thus two sets of thresholds $\{R_{th}^{1,2}, R_{th}^{2,3}\}$ and $\{n_{th}^{1,2}, n_{th}^{2,3}\}$. We show the three different cases for a given time point in a trajectory, classified by their mobility from left to right as $s_1$, $s_2$ and $s_3$.

For correct segmentation, one needs a measure to assess the performance of specific segmentation parameters (for example, the set of thresholds $\{R_{th}\}$ and $\{n_{th}\}$) according to their predictive power. For this, we turn to the property of Gaussianity of step-size distributions. Random walks originating from a distinct process effectively yield a Gaussian distribution of step sizes. The Gaussian distribution appears after many steps of the random walk by virtue of the central limit theorem[43], regardless of the original distribution of step sizes of the underlying process. However, the step-size distribution of our measured trajectories results in a highly non-Gaussian distribution, as evident from the deviation of the step-size distribution (blue circles) from the best Gaussian fit (red line) in Fig. 3c. Step-size distributions of SPT

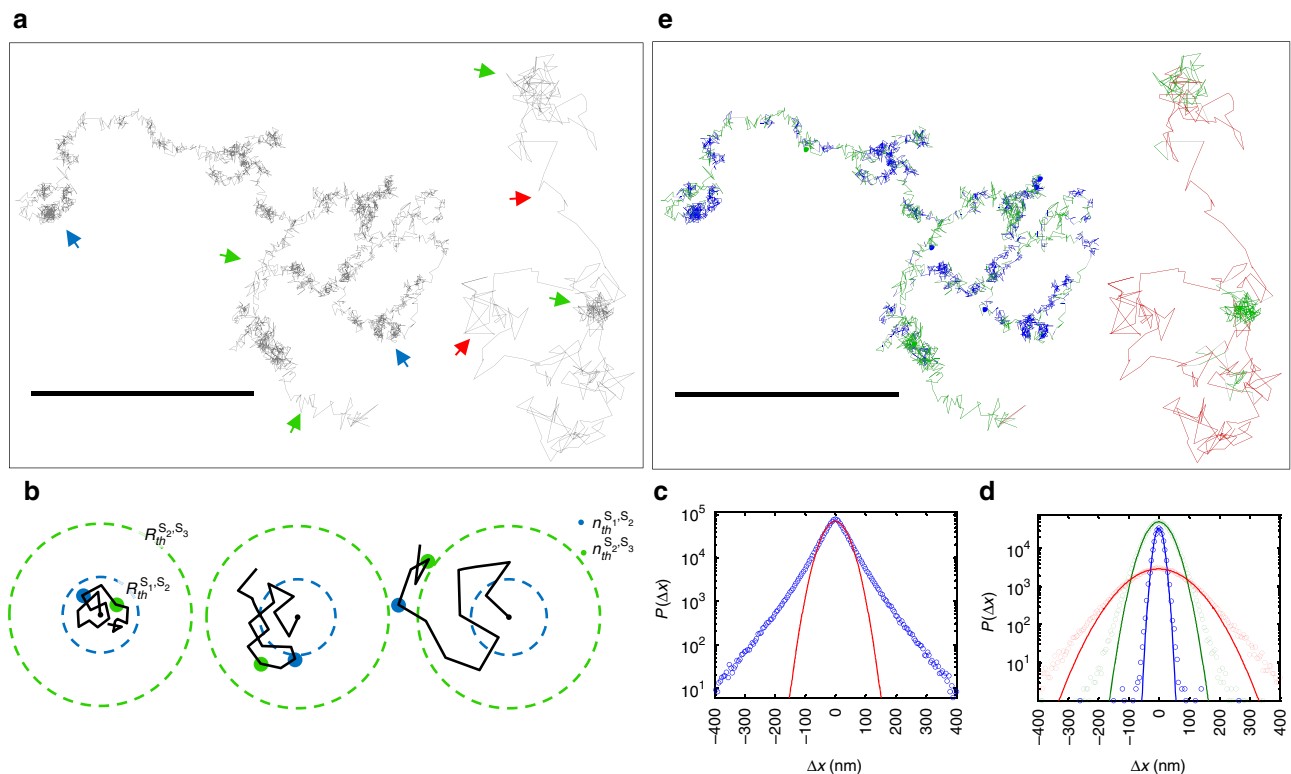

**Figure 3 | Heterogeneity of subdiffusive motion at single trajectory level.** (**a**) Representative trajectories exhibiting dynamically changing diffusive modes. The denser areas are ones where the particles dwell for longer durations. Pointed out in blue, green and red arrows are areas which are highly dense, moderately dense and relatively spread out. These observations suggest the need for segmentation of transient mobility states. Scale bar, 1 μm. (**b**) A schematic representation of the CER segmentation process for a model of three mobility states. Blue and green dashed circles and blue and green dots are the escape radii thresholds $R_{th}^{s_1,s_2}$ and $R_{th}^{s_2,s_3}$ and the time steps thresholds $n_{th}^{s_1,s_2}$ and $n_{th}^{s_2,s_3}$, which distinguish between states 1,2 and 2,3, respectively. In this example, we chose $n_{th}^{s_1,s_2} = 9$ and $n_{th}^{s_2,s_3} = 11$. The leftmost example shows a case where the particle did not escape from $R_{th}^{s_1,s_2}$ in $n_{th}^{s_1,s_2}$ time steps and thus the tested time point is classified as $s_1$, and accordingly, the middle and rightmost examples are classified as $s_2$ and $s_3$. (**c**) Step-size distribution of 6,410 mobile trajectories 50–5,000 time steps long from 30 cells. The best Gaussian fit is presented as a red line. (**d**) Step-size distribution of segmented populations of three states, separated by their high, medium and low mobility (red, green and blue circles, respectively). Solid lines are best Gaussian fits of the segmented subpopulations. (**e**) The representative trajectories from **a** after segmentation. Segments are coloured blue, green and red according to their mobility states $s_1, s_2, s_3$. Scale bar, 1 μm.

data are often non-Gaussian, which may arise from tracer or environmental heterogeneity[21,24,44–46]. Step sizes may be inherently non-Gaussian in cases of fluctuating environments on similar timescales as the random walk itself[46]. Still, at long timescales even such cases would appear as Gaussian[44,46]. Thus, regardless of the method by which we segment trajectories, we expect that the detected distribution of step sizes should be divided into multiple Gaussian components, each representing a separate underlying random-walk process. Therefore, an efficient segmentation process can be evaluated based on the extent of data it is able to account for using a minimal number of such components. Note that in vbSPT, the maximum evidence criterion does not consider the statistics of the step-size distribution for the robustness of the state.

With this consideration, we scan a large range of threshold sets $\{R_{th}\}$ and $\{n_{th}\}$ for the CER method. We then test for each set the extent of step-size Gaussianity and mixing of the resulting subpopulations recovered by the segmentation process. Specifically, we fit a finite mixture of $k$ Gaussian components to each subpopulation, assumed by a model with an arbitrary number of subpopulations $k$:

$$P^j(\Delta\vec{r}) = \sum_{i=1}^{k} a_i^j \frac{1}{2\pi\sigma_i^2} e^{-\frac{\Delta r^2}{2\sigma_i^2}} \quad (8)$$

where $P^j(\Delta\vec{r})$ is the step-size distribution for subpopulation $j$, $a_i^j$ is the weight of each Gaussian component $i$ in subpopulation $j$ and $\sigma_i^2$ is the variance of each Gaussian component. Thus, $a_i^j$ is a matrix element describing the weight of the $i$th Gaussian component of the $j$th subpopulation. According to our model, we require that $\sum_{i=1}^{k} a_i^j = 1$ and that $0 \le a_i^j \le 1$. We also constrain each $\sigma_i$ to be constant over all $k$ subpopulation fits since we assume that the underlying Gaussian components are constant and only their weights in each subpopulation ($a_i^{j's}$) are changing. This assumption is justified since we identify each Gaussian component with a corresponding, distinguishable physical mechanism. The segmentation efficiency is measured by minimization of the off-diagonal $a_{i\ne j}^j = 0$. Thus, an ideal segmentation would result in $a_{i=j}^j = 1$ and $a_{i\ne j}^j = 0$ for all $i,j$ (see 'Methods' section and Supplementary Fig. 5 for examples and further details).

It is *a priori* unclear how many states are present in the data (or moreover, if states can be regarded as discrete in the first place). Yet, any segmentation effort needs a model with a certain number of states to account for. Thus, we consider models that grow in their number of states, and thus in their complexity. These competing models should be compared according to their ability to account for the data. However, simpler models with the same

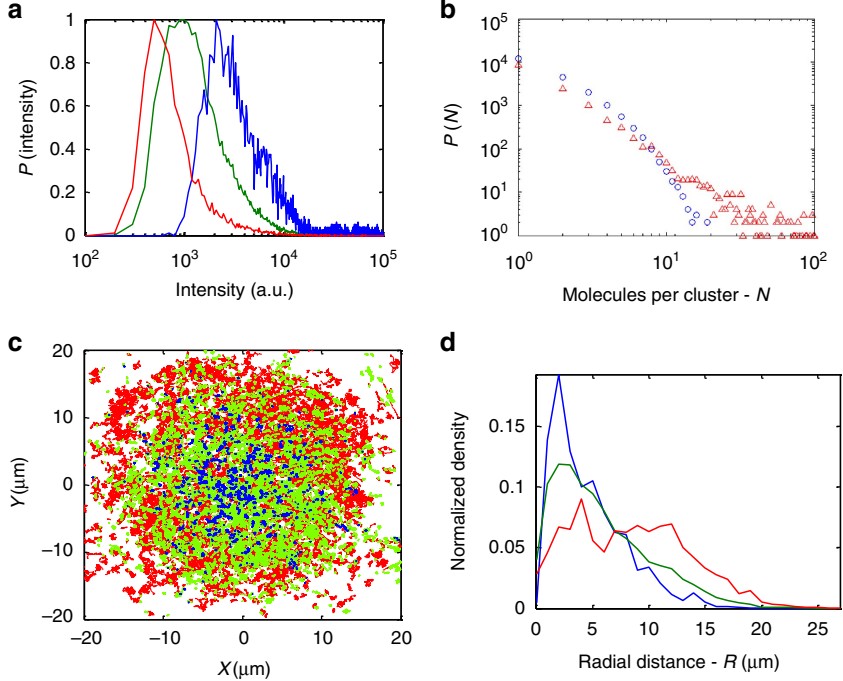

**Figure 4 | Characterization of the three different mobility subpopulations.** (**a**) Normalized distribution of fluorescence intensity per segment for each mobility subpopulation. The intensity value is taken from the mean of the first 10 time steps of each trajectory segment to minimize photobleaching effects. The values are given for each localization by the ThunderStorm algorithm[64]. High, medium and low mobilities are shown in red, green and blue, respectively. (**b**) Cluster size distribution calculated from a sample cell imaged by PALM. Data points (red triangles) and results for randomly distributed points (blue circles) are shown. The calculation was performed on an area of $10 \times 10 \, \mu m$ in the cell centre. (**c**) Montage image of trajectory segments from 30 different cells. The centre of mass of all localizations for each cell was used for aligning the different cell images. Colour coding as in **a**. (**d**) Radial density distribution of segment subpopulations with different mobility states. Colour coding as in **a**.

predictive power are superior to more complex ones due to Occam's razor. States that converge to similar underlying mechanisms and mobility characteristics may be assumed redundant and thus, make the suggested model more robust. Hence, a useful approach is to start with a low number of states, and then increase their number until redundancy of one of the states is achieved. Note that this redundancy can be inferred only after a thorough characterization of the mobility states and their underlying mechanisms.

Next, we consider models with an increasing number of states from two to four. Our analyses show that the three states model accounts best for the data and its underlying subdiffusion mechanisms. We first describe the complete set of statistics for the three states model and, only then, fully discuss the inadequacy of the two and four states models at the end of the text. Briefly, using a three states model, our analysis identifies three distinct subpopulations, which we term as 'low', 'medium' and 'high' mobilities. In contrast, the two states model misses a subpopulation with the distinct mobility properties of the 'high' mobility state and its unique underlying mechanism. With the four states model, we find that the 'high' mobility subpopulation has the same characteristics of the 'high' mobility subpopulation found for the three states model. However, the 'medium' and 'low' mobility states become redundant and separate into three subpopulations with essentially the same underlying mechanisms and overlapping mobility characteristics (see 'Discussion' section).

Returning to the three states model, Fig. 3d shows the step-size distributions for each of the subpopulations detected by the CER method. Each single Gaussian accounts for the majority ($>95\%$) of its fitted data. Figure 3e presents again the representative trajectories shown in Fig. 3a after segmentation into three mobility states.

At this point, we would like to clarify the relation between different terms used in the text that are invoked through our segmentation and classification processes. Through these processes, segment subpopulations are uniquely assigned to specific mobility states. Thus, we will now interchangeably use the terms 'mobility states', 'mobility subpopulations' or simply 'subpopulations' to describe the segments of common mobility characteristics (being either 'low', 'medium' or 'high'). Moreover, we note that through the segmentation process, a trajectory of a single particle may be divided into several segments that match different mobility states and may thus be assigned to several segment subpopulations.

**Characteristics of the different subpopulations.** Once we segment all trajectories into three subpopulations, we begin our analyses of each subpopulation to determine its underlying subdiffusion mechanism. Starting with general characteristics of the subpopulations, Fig. 4a shows the normalized distributions of fluorescence intensity of the three subpopulations. This is not a direct characterization of single particles, but rather of their segmented trajectories that might belong to different subpopulations. However, for physically stable particles (that is, they do not disintegrate or assemble over the typical trajectory duration (Supplementary Fig. 1c)), we can infer on average the particle fluorescence intensity distributions from this analysis. Also, although the fluorescence intensity is not a precise measure of the number of proteins in individual particles, it can give a rough estimate of this number. This lack of precision is due to the possible existence of multiple emitters per tracer. For instance, single antibodies may carry multiple fluorophores ($\sim 3$ in our case; see 'Methods' section), multiple antibodies may target

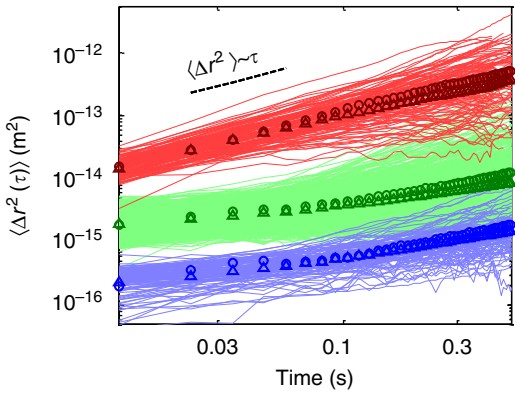

**Figure 5 | MSDs for the different mobility subpopulations.** A log–log plot of MSDs of the three different mobility subpopulations. Only 10% of the trajectories are shown for clarity. Top group (red)—high mobility, middle group (green)—medium mobility, bottom group (blue)—low mobility. Circles and triangles are, respectively, the ensemble and mean of the time averages of each group. The corresponding subdiffusive $\alpha$ and $K_\alpha$ values are shown in Table 1.

| | Low | Med | High |
|---|---|---|---|
| **Table 1 | Subdiffusive power $\alpha$ and $K_\alpha$ [m²s⁻ᵅ] for the three mobility states found by the CER method.** | | | |
| $\alpha_e$ | 0.67(6) | 0.55(1) | 0.91(2) |
| $\alpha_t$ | 0.55(1) | 0.40(2) | 0.83(2) |
| $\alpha_{MME}$ | 0.65(1) | 0.59(1) | 0.96(2) |
| $K_{\alpha,e}$ | $3.0(8) \times 10^{-15}$ | $1.7(2) \times 10^{-14}$ | $0.9(1) \times 10^{-12}$ |
| $K_{\alpha,t}$ | $2.1(5) \times 10^{-15}$ | $1.0(1) \times 10^{-14}$ | $0.6(1) \times 10^{-12}$ |
| $K_{\alpha,MME}$ | $6(1) \times 10^{-15}$ | $3.4(5) \times 10^{-14}$ | $1.2(2) \times 10^{-12}$ |

The ensemble, time average and MME values are calculated by the mean of the values from all threshold sets that meet the guidelines in the CER segmentation process. 625 unique threshold sets are computed and 83 meet the guidelines (see 'Methods' section for further details). The s.d. of the value is shown as an error of the last digit in parenthesis. Fitting over multiple time windows of the MSD results in a larger effective error of ∼0.04 for the $\alpha$ values, while the $K_\alpha$ errors are comparable to those shown.

an individual protein or gp41 proteins may self-cluster[32,47,48]. In cases of multiple targeting of antibodies and self-clustering, the size of the tracer increases and thus its mobility decreases. We find that segments belonging to brighter particles are of a lower mobility state on average (Fig. 4a).

To better characterize the size distribution of particles, we turn to PALM, which is enabled by the unique construct we use for SPT. In contrast to immunofluorescence, PALM can visualize individually all of the genetically encoded gp41 proteins in each cluster with minimal artifacts and with a resolution down to ∼25 nm (ref. 49). The detected molecules can then be clustered by assigning proximal molecules beneath a specific distance threshold to the same cluster. We chose a distance threshold of 40 nm that accounts for the resolution limit in the localization of two proximal molecules (see 'Methods' section for further details). Figure 4b shows the cluster size distribution of molecules from a sample cell in logarithmic scale. We compare this size distribution to the one from a random scatter (that is, a Poisson process) of points in the given area. We find that the number of molecules per cluster is skewed towards clusters as small as dimers and trimers, but has a long tail compared to the Poisson distribution. This distribution indicates the existence of larger clusters than expected by a random scatter and high heterogeneity in their size.

Figure 4c,d shows the spatial distribution of trajectory segments associated with the three mobility states. We find that the higher the mobility of a segment, the farther its corresponding particle can distribute from the centre of the cell footprint on average. Thus, lower mobility states tend to be correlated with aggregation of proteins and tend to be enriched at the centre of the cell footprint. These observations strengthen the validity of our segmentation process since they capture independent characteristics of the tracer particles without *a priori* assumptions.

We now return to the MSD statistics for further characterization of the subpopulations. Figure 5 shows the ensemble and time averages of the trajectory segments of subpopulations associated with each of the mobility states. The intersection of the MSD curves with the *y* axis shows that the three mobility states are highly separable and have variable diffusion coefficient values, $K_\alpha$. Note that our segmentation process excludes segments shorter than 50 frames. Thus trajectories with non-distinct mobility states may be depleted during this process and result in more separable $K_\alpha$ values. The subdiffusive powers $\alpha$ and the diffusion

coefficients $K_\alpha$ obtained for each of the three mobility states are shown in Table 1. We find that all three mobility states are subdiffusive with non-negligible differences between the time and ensemble values. We account for these differences in the discussion. The low and medium mobility states exhibit a similar range of $\alpha$ values (with somewhat higher $\alpha$ values for the low mobility state), but differ strongly in their diffusion coefficients, $K_\alpha$. The high mobility state is only slightly subdiffusive and is substantially different from the other subpopulations in its mobility characteristics $\alpha$ and $K_\alpha$. Importantly, the vbSPT segmentation method yields three subpopulations with similar, yet less separable, $K_\alpha$ and $\alpha$ values compared with the CER method (Supplementary Table 1).

**Assigning mechanisms.** To further study each subpopulation, we calculate their velocity autocorrelation (VAC) functions:

$$\frac{C_v^\delta(\tau)}{C_v^\delta(0)} = \frac{\langle \vec{v}(t) \cdot \vec{v}(t+\tau) \rangle}{\langle \vec{v}(t) \cdot \vec{v}(t) \rangle} \qquad (9)$$

where

$$\vec{v}(t) = \frac{1}{\delta}\left(\vec{r}(t+\delta) - \vec{r}(t)\right) \qquad (10)$$

is the particle velocity at time $t$ calculated over a time period $\delta$. The VAC is computed separately for each trajectory and then averaged over the subpopulation ensemble. The shape of the VAC can separate well CTRW from confined motion and ergodic processes[11,29,30,50], but cannot distinguish between fBM and RWF[18].

Figure 6 shows the VAC functions of the three subpopulations for a range of $\tau$ values and changing time periods $\delta$. For both fBM and RWF, we expect to find anti-persistent correlations at initial $\tau$ values that decay to 0 after some time. These negative correlations should appear also for longer timescales $\delta$. For CTRW, we expect to find no correlations at all, since the random walker can alter its direction of movement between trapping events. For a confined random walk, we expect to find no correlations at short timescales $\delta$, but negative correlations at longer timescales.

The low and medium mobility states are similar and both exhibit negative correlations that are insensitive to the time periods $\delta$ over a wide range of values. This feature confirms our initial ruling out of CTRW as a possible underlying mechanism and points to either fBM or RWF as the dominating mechanisms of subdiffusion. In contrast, the high mobility state shows markedly different features than the low and medium mobility subpopulations. At the initial time period $\delta$, on the order of our time resolution of 11 ms, it shows no (or negligible) correlations. It then shows growing negative correlations for longer $\delta$ values. This behaviour, together with the $\alpha$ value being close to 1,

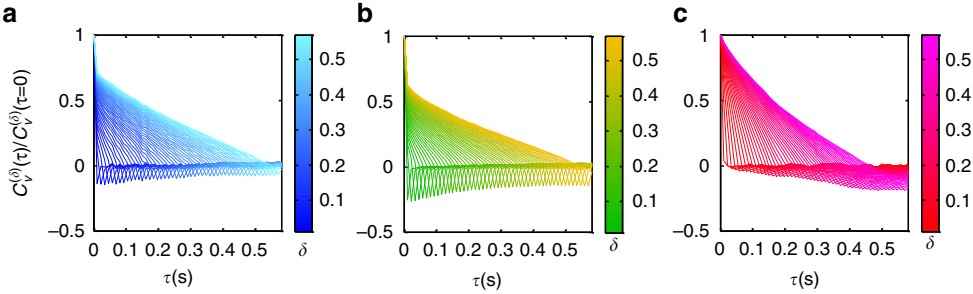

**Figure 6 | VACs.** VAC functions for the three different mobility subpopulations—(**a**) low, (**b**) medium and (**c**) high. Each plot consists of 50 different timescales $\delta(11.6\,\mathrm{ms} - 0.58\,\mathrm{s})$ over which the velocity is measured. Each timescale is plotted in a gradient of the corresponding colour for clarity (blue, green, red for low, medium and high, respectively). Localization errors were taken into account as suggested in ref. 29.

indicates that the motion of the high mobility subpopulation is mostly Brownian, yet confined[29]. The appearance of growing negative correlations with $\delta$ immediately after the initial $\delta$ value can be understood from the timescale $\overline{\Delta t}$ needed for a particle to travel the mean step size of the high mobility subpopulation $\overline{\Delta r} \cong 71\,\mathrm{nm}$. This value can be estimated by $\overline{\Delta t} \sim \left(\frac{\overline{\Delta r}^2}{K_\alpha}\right)^{1/\alpha} \approx 3\,\mathrm{ms}$, which is shorter than our time resolution. Thus, we conclude that for small $\delta$ values only a small fraction of the particles is affected by the boundary, whereas at growing $\delta$ values, an increasing fraction of the particles is affected by it. We refer to this as weak confinement of the high mobility subpopulation. Conversely, strong confinement would result in significant negative autocorrelations at the initial time period $\delta$. Furthermore, we note that our analyses are limited by the trajectory lifetimes to $\sim0.6\,\mathrm{s}$. This time frame is likely too short for the trajectory segments to regain apparent Brownian MSDs (Fig. 5) and null velocity autocorrelations at long time periods (Fig. 6c).

We now focus on the study of the underlying mechanisms for the low and medium subpopulations, since for these subpopulations, the VAC alone can not distinguish between fBM and RWF[18]. We first turn to PALM imaging, as it enables the spatial mapping of available sites for the particles and, consequently, the characterization of the fractal dimension of the cell footprint (related to RWF). We assume the particles are homogeneously spread over the structure imaged by PALM, and thus is effectively captured by it. We calculate the fractal Hausdorff dimension of the gp41 spatial distribution across the PM using the box-counting method[51] (Fig. 7a,b). We find a fractal dimension $d_f = 1.84 \pm 0.07$. Notably, this calculation excludes the lowest scales, where the resolution limit is bounding, and the largest scales, where the footprint of the cell becomes significant. The observed fractal-like structure may not be the only mechanism that affects the particles mobility. Thus, we turn below to two complementary statistical tests of the PDF Gaussianity of the SPT trajectories, namely moment ratios analysis[11,28,52] and the growing sphere analysis[11,28].

The moment ratio analysis tests the extent of Gaussianity of the regular (MSD) and mean maximal excursion (MME) by calculating the regular and MME moment ratios. Note that earlier we tested the step-size distribution for Gaussianity while here, we test the Gaussianity of the PDF of the diffusion process. We compute the time-averaged MME for each trajectory segment as:

$$\langle \Delta r^k_{\max}(\tau \cdot \Delta t) \rangle_{\text{time}} = \frac{1}{T-\tau} \sum_{i=1}^{T-\tau} \left( \max_{0 \le \tau' \le \tau} \{\vec{r}_{\tau'+i} - \vec{r}_i\} \right)^k \quad (11)$$

where $k$ is the moment and $T, \tau$ and $\Delta t$ are defined as in equation (3). We provide the MME values for $\alpha$ and $K_\alpha$ in

Table 1. Note that the MME values for $\alpha$ are higher than the ensemble and time-averaged values. This is expected for an fBM process, but cannot be regarded as conclusive evidence for it[28]. The regular and MME moment ratios are defined as:

$$\frac{\langle \Delta r^4(\tau) \rangle}{\langle \Delta r^2(\tau) \rangle^2}, \frac{\langle \Delta r^4_{\max}(\tau) \rangle}{\langle \Delta r^2_{\max}(\tau) \rangle^2} \quad (12)$$

and are used as measures for the PDF Gaussianity. These moment ratios should have distinct asymptotic values (that is, for $\tau \gg 1$) for fBM and RWF processes. These asymptotic values depend on the subdiffusive power $\alpha$ and, in the case of RWF, also on the fractal dimension $d_f$ (ref. 28). For an fBM process, the regular asymptotic moment ratio converges to the value 2.0. There is no exact analytical solution for the MME moment ratio, but an approximation[28]:

$$\frac{\langle \Delta r^4_{\max}(\tau) \rangle}{\langle \Delta r^2_{\max}(\tau) \rangle^2} \approx (1.05 \pm 0.01)\left(\frac{\alpha}{2}\right)^{1.42 \pm 0.01} + (1.10 \pm 0.01) \quad (13)$$

Also, in the case of RWF, there is an approximated solution, which can be numerically computed for any choice of $d_f$ and $\alpha$ (ref. 28).

Figure 7c shows the regular (crosses) and MME (triangles) moment ratios for the low (blue) and medium (green) mobility subpopulations. We mark the expected values for fBM and RWF processes for the low and medium subpopulations for the regular and MME moment ratios (overall, seven lines in the figure). Our calculated moment ratios show asymptotic values, which for the MME moment ratios fit slightly better the RWF model, yet hard to distinguish from fBM. More distinctly, the asymptotic values of the regular moment ratios converge to the RWF expected values. This supports the possibility of an RWF process as an underlying mechanism.

The second statistical test is the growing sphere analysis[11,28]. In this method, we calculate the probability of a particle to be within a growing sphere with a radius increasing as $\propto r_0 t^{\alpha/2}$:

$$P\left(r \le r_0 t^{\frac{\alpha}{2}}\right) \approx \frac{1}{N(t)} \sum_i^{N(t)} \Theta\left(r_i(t) - r_0 t^{\alpha/2}\right) \quad (14)$$

where $\Theta(r)$ is the Heaviside function (that is, 1 for $r \ge 0$ and 0 for $r < 0$) and $N(t)$ is the number of particles present at time $t$. For an fBM process, this ratio should be constant, while for an RWF it should scale as $\propto t^{\alpha(d - d_f)/2}$ (ref. 28), thus showing a power law growth with $t$. Figure 7d shows the growing sphere analysis of the low and medium subpopulations. We find that the values for the growing sphere analysis are constant, with an exception of the shortest timescales, which is found to be a property of the growing sphere analysis as demonstrated by simulated particles (black dash-dotted line) in Fig. 7d. This indicates fBM as an

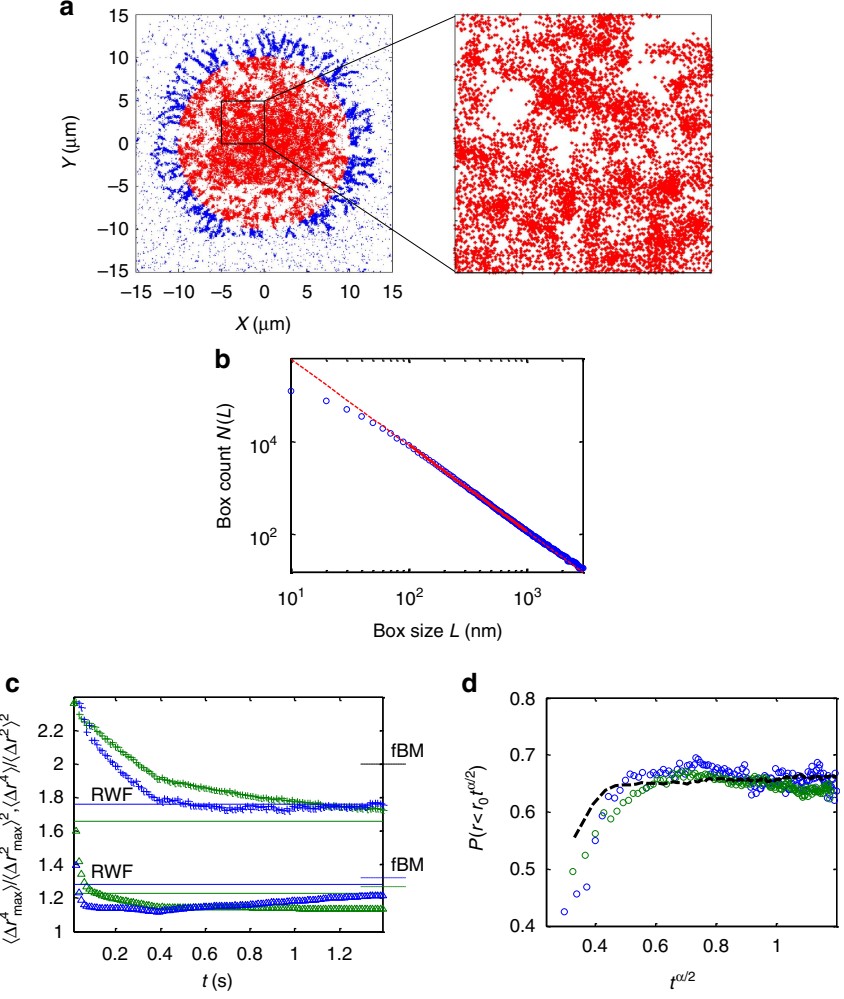

**Figure 7 | Imaging and analyses for distinguishing RWF from fBM.** (**a**) PALM image of a representative cell showing a fractal-like structure with $d_f \approx 1.87$. The fractal dimension is measured for the red points, which are the majority of the cell footprint. (**b**) Log–log plot of the box counting algorithm results for the PALM image in **a**. The red line is a linear fit to the data points in the range 10–1,000, which is emphasized by the solid part of the line. The fractal dimension is the slope of the fit. (**c**) Regular (MSD) moment ratios (crosses) and MME moment ratios (triangles) for low (blue) and medium (green) mobility states. The expected values are marked for both fBM (short marks on the right axis) and RWF (long horizontal lines) processes. The expected values come in pairs for the low (blue) and medium (green) mobility subpopulations according to their respective $\alpha_e$ values from Table 1. The expected values are: fBM regular moments at 2.0; fBM MME moments at 1.32 and 1.27; RWF regular moments at 1.76 and 1.66; RWF MME moments at 1.28 and 1.2. (**d**) Growing sphere analysis for low (blue circles) and medium (green circles) mobility states. A simulated result of $2 \times 10^4$ Brownian particles is added for comparison (black dashed line).

added underlying mechanism to RWF for the subdiffusive mobility of both subpopulations.

**Cell activation effects on subdiffusion mechanisms.** T-cell activation results in significant changes in its morphology as it adheres to APCs. It is unclear how such changes affect protein mobility at the cell PM[53]. To study the effect of cell activation, we now turn to results from imaging of cells under non-activating conditions, as the cells adhere to coverslips that do not stimulate the TCR (see 'Methods' section). Figure 8 shows the summary of the different tests for these cells using the three states model. As for the activated cells, we find both dynamic and static heterogeneity in the MSDs, thus we follow the same segmentation procedure. After segmentation, using the CER approach, we observe similar overall characteristics: (1) the segments can be separated into subpopulations of low, medium and high mobility ($\alpha_e = 0.56 \pm 0.08$, $0.49 \pm 0.04$ and $0.92 \pm 0.02$, respectively). (2) These subpopulations are distributed differentially in the cell footprint as the high mobility subpopulation is evenly

distributed and extend to the periphery of the cell, while the lower mobility subpopulations localize to the centre (Fig. 8a,b). (3) The time and ensemble MSDs are consistent with ergodicity of all underlying processes (Fig. 8c). (4) The VAC indicates fBM or RWF processes as the underlying mechanisms of the low and medium mobility subpopulations. The VAC of the high mobility subpopulation shows again confinement, yet a larger negative correlation at short time periods (Fig. 8d–f). (5) The moment ratios for the low and medium mobility subpopulations correlate better with fBM as their regular moments converge to a value of $\approx 2$ and the MME ratios converge to their expected values (Fig. 8g). (6) The growing sphere analysis also points to fBM, in agreement with the moment ratio analysis (Fig. 8h). Thus, we conclude that without cell activation, the PM is more viscoelastic and less fractal-like as indicated by the mobility properties of the low and medium subpopulations. Still, the three distinct mobility subpopulations are common to activated and non-activated cells, under our measurement conditions.

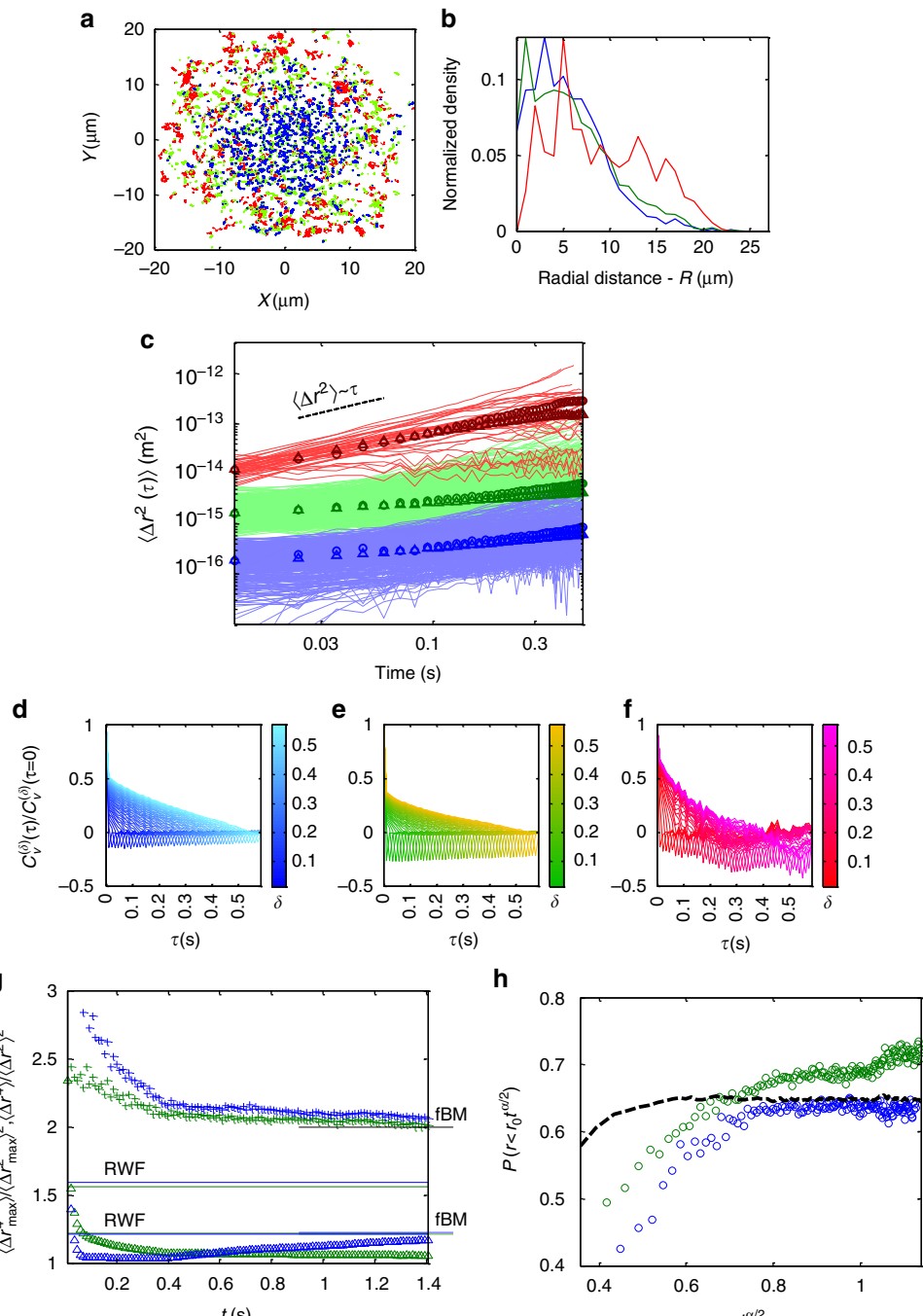

**Figure 8 | The subdiffusion mechanisms in non-activated cells.** Repetition of statistical tests for non-activated cells. All panels are the same as the corresponding panels in Figs 4–7. (**a**) Montage image of trajectories from 30 different cells. (**b**) Radial density distribution of mobility states. (**c**) A log–log plot of MSDs of the three different mobility subpopulations. (**d**–**f**) VAC functions for the three different mobility subpopulations. (**g**) Regular (MSD) moment ratios (crosses) and MME moment ratios (triangles). The expected values are marked for both fBM (short marks on the right axis) and RWF (long horizontal lines) processes. The expected values are shown in pairs for the low (blue) and medium (green) mobility subpopulations according to their respective $\alpha_e$ values. (**h**) Growing sphere analysis. Colouring is blue, green and red for mobility states $s_{1,2,3}$ in ascending order of mobility.

## Discussion

In this study, we detect heterogeneity of protein mobility at the PM of activated and non-activated T cells. This heterogeneity appears to be both static and dynamic. We observe polydispersity of particles containing a transmembrane protein derived from gp41 and dynamic heterogeneity in which particles may change their diffusion state during their motion. After segmentation of trajectories into subpopulations with distinct mobility states, we notice a third heterogeneity in the spatial distribution of the segment subpopulations, where more mobile trajectories can distribute farther from the centre of the cell footprint. This complexity, in the form of multiple levels of heterogeneity, is likely a common feature of biological samples and is generally difficult to capture and characterize on the whole ensemble level or with small data sets of particle trajectories. Such a complexity is often simplified by using, artificial and *in vitro* systems

including vesicles and lipid bilayers[38], model fluids, computer simulations and so on.[7,40,54]. However, multiple studies have characterized macromolecular diffusion within the PM of live cells[7,40,54]. A few recent studies have detected underlying heterogeneity in SPT results and suggested unifying models to explain their findings[21–26]. However, these studies typically do not attempt to separate the data into subpopulations with distinct mobility states. Thus, they cannot resolve potentially distinct underlying mechanism acting differentially on different particle subpopulations and during the diffusion of individual particles.

The dynamic heterogeneity of single trajectories and the non-Gaussianity of their step-size distribution led us to apply segmentation of trajectories according to local variations in particle mobility. We use two complementary segmentation approaches: vbSPT[25] and our modified CER approach. Alternative segmentation approaches may be applied to SPT data, including temporal sliding window, Bayesian algorithms, HMM or supervised classification[40]. Each approach incorporates dynamic variables that serve for classification. However, the target of classification may be subjective. Here we offer the property of Gaussianity of the step-size distribution as a universal classifying target for effective segmentation of distinct subpopulations. This approach is applicable to a wide range of mixed subdiffusive or Brownian random walks, however, excluding systems of fluctuating environments on similar timescales as the random walk itself[46]. Thus, an optimal classifier for mixed random walks may include a robust classifier (for example, vbSPT) with a consideration of the possibility of anomalous diffusion of the subpopulations and with a requirement for the Gaussianity of their individual step-size distributions.

Notably, our analyses rely on a first robust step of rejecting immobile species based on their radius of gyration $R_g$ and mean step size $\langle |\Delta r| \rangle$. Such rejection is important since this immobile fraction often dominates the population of single trajectories[21,22], and may complicate attempts to distinguish the step-size Gaussianity of distinct subpopulations.

In our segmentation process, we consider models with an increasing number of mobility states, from two to four. A two states model fails to separate the medium and high mobility subpopulations as it fails to capture the VAC properties of the high mobility state, while showing dominating features of the medium mobility state (Supplementary Fig. 3, compare panels (d,e) to Fig. 6). Next, we consider a three states model. This model accounts for >95% of the data of each subpopulation. As expected, the subpopulations that correspond to the three mobility states differ in their diffusion coefficient $K_\alpha$. The subpopulation corresponding to the high mobility state undergoes weak confinement as its dominating underlying mechanism. Surprisingly, the two subpopulations with the lower mobility states are found to share a common underlying mechanism as they have similar mobility characteristics, including comparable $\alpha$ values, and similar VAC curves, moment ratios, growing sphere analysis and spatial distribution. These statistical tests indicate either RWF, fBM or their combination as the underlying mechanism for subdiffusion. Having two subpopulations of low and medium mobilities that differ predominantly in their $K_\alpha$ values, but otherwise share multiple statistical characteristics, strongly suggests that they belong to a common subpopulation of particles. The particles in this subpopulation are affected by the environment through the same mechanism and only differ in their size-dependent mobility. To confirm this conclusion, we test the data using a four states model (Supplementary Fig. 4) and find, yet again, a highly mobile subpopulation with the same characteristics indicating weak confinement. The remaining three lower mobility states have differing $K_\alpha$ values, but otherwise, a common underlying mechanism (Supplementary Table 2).

For all subpopulations, we notice non-negligible differences between the mobility values $\alpha$ and $K_\alpha$ of the time and ensemble averages (for example, Table 1). This opens the possibility of the involvement of non-ergodic mechanisms (for example, CTRW). A previous study[20] predicts specific dependencies of the ensemble and time MSDs on the subordination of a CTRW on a fractal lattice. This study suggests a relation of $\alpha_e = \alpha_{CTRW} \times \alpha_{RWF}$ and $\alpha_t = 1 - \alpha_{CTRW} + \alpha_{CTRW} \times \alpha_{RWF}$. This implies that $\alpha_{CTRW} = 1 + \alpha_e - \alpha_t$. Since $\alpha_t < \alpha_e$ in all of our results, this yields erroneous values of $\alpha_{CTRW} > 1.0$. An alternative explanation is the general effect of the weak confinement, which we find for the high mobility subpopulation, on all subpopulations. Indeed, weak confinement can generate inequivalent time and ensemble averages in ergodic systems[55]. To test this explanation, we simulate combinations of weak confinement and either Brownian motion or fBM (see 'Methods' section). We find that in both cases, the confinement leads consistently to $\alpha_t < \alpha_e$ with a difference of ∼5–20%, depending on the relative radius of confinement, simulated $\alpha$, localization uncertainty and MSD fitting accuracy (for example, Supplementary Fig. 6). This explanation also holds for the small observed difference in the $\alpha$ values of the ensemble and time averages before the segmentation process (Fig. 2). Thus, we conclude that weak confinement can sufficiently account for the inequivalence in the $\alpha_e$ and $\alpha_t$ values in all presented cases. We also observe that consistently, $K_{(\alpha,t)} < K_{(\alpha,e)}$. This is understandable since the MSD of the time average is computed as the geometric average of all trajectories, while the MSD of the ensemble is an arithmetic average. Therefore, $K_{(\alpha,t)}$ consistently underestimates the true $K_\alpha$.

It is difficult to distinguish between RWF and fBM as the underlying mechanisms of the 'low–medium' mobility subpopulation. The moment ratios test and the PM fractal dimension found by PALM indicate RWF as the dominating mechanism. However, the asymptotic behaviour of the growing sphere analysis (Fig. 7d) indicates a possible fBM process, which cannot be excluded by VAC[11]. Thus, our data suggest the existence of both mechanisms acting simultaneously. Statistical tests have been suggested and applied for resolving the subordination of subdiffusion mechanisms[11,19]. However, these tests have been applied to the separation of ergodic (for example, fBM or RWF) from non-ergodic (for example, CTRW) processes, while clear guidelines and concrete examples for the separation of mixed ergodic processes seem missing.

Both fBM and RWF have been associated with molecular crowding[56,57]. Structural patterning of the PM may additionally cause RWF, while a viscoelastic mesh may give rise to fBM[58,59]. We further show that these mechanisms depend on cell activation as RWF is dominant for activated cells, while fBM is dominant for non-activated cells. Indeed, cell activation has been shown to cause global changes in the morphology of the PM and the cytoskeleton[2,3]. These changes include coalescence of signalling proteins into micro-clusters, polymerization of cytoskeletal proteins and tighter adhesion with the APC and spreading. These changes might alter the diffusive motion of proteins at the PM. For instance, signalling proteins might get trapped in clusters[39] or in actin corrals[58,60], experience diffusion barriers[61], while the PM might change its three-dimensional patterning[62]. Here we use a relatively inert transmembrane protein, namely the gp41 transmembrane domain with a mutation that suppresses its interactions with specific lipids[32] and cellular proteins at the PM. However, specific interactions of this protein with the PM environment cannot be completely ruled out. Naturally, other transmembrane proteins in the cell may undergo significant and specific interactions to facilitate their action, and thus may exhibit different underlying subdiffusive

mechanisms than are found here (for example, non-ergodic processes)[40].

Taken together, our results and statistical tests demonstrate the simultaneous existence of multiple subdiffusion processes, namely confinement, RWF and fBM at the PM of cells. These mechanisms act differentially on particles depending on their size and may change their dominance under varying cell activation conditions. Thus, our results shed new light on the multiple mechanisms that govern protein mobility at the PM of activated and non-activated T cells. More generally, our approach and results demonstrate how one can methodologically resolve mixed mechanisms of subdiffusion that act simultaneously on a polydispersed sample within the complex medium of the cell PM. This approach can also serve as a framework for resolving mixed mechanisms in a wide range of random walks of arbitrary nature.

## Methods

**Sample preparation.** Gp41-PAGFP[63] was generated from previously published constructs of gp41 (Δ1 mCRAC gp41 (ref. 32)). DNA was introduced into E6.1 Jurkat T cells (a kind gift from the Samelson lab at the NIH) using the Neon electroporation system (Invitrogen). Cells were next incubated at 37 °C for 48 h. Since PAGFP is not stable or bright enough for SPT, we used it as a specific target for immunofluorescence labelling. For labelling, 0.5 μg anti-GFP, rabbit polyclonal antibody, Alexa Fluor 594 (AF594) conjugate (Life technologies Cat. A21312 Lot 1476604) was added to $5 \times 10^5$ cells suspended in FACS buffer for 30 min on ice. According to vendor specifications, there are three AF594 fluorophores bound to each antibody molecule. The cells were washed three times with PBS and suspended in an imaging buffer. Cells were dropped onto a coated coverslip for live cell imaging. For fixed cell imaging, the cells were first incubated for 20 min and then PFA was added at a ratio of 3/5 for cell fixation over another 30 min (in an incubator) before washing and imaging.

For imaging, we used coverslips (Ibidi μ-8-well glass bottom No. 1.5H) coated with 0.01% poly-L-lysine (Sigma) and αCD3ε (UCHT1; BD Biosciences) or αCD45 (BD Biosciences) antibodies at a concentration of 10 μg ml$^{-1}$. αCD3ε coating robustly stimulates the TCR and leads to T-cell activation and spreading. In contrast, the αCD45 coating does not stimulate the TCR and serve for T-cell adherence to the coverslip without its TCR-dependent activation.

**Imaging.** Imaging was performed on a TIRF microscope (Nikon) with a CFI Apo TIRF × 100 oil objective (NA 1.49, WD 0.12 mm). Imaging in TIRF mode served to visualize molecules at the PM of spreading cells in close proximity to the coverslip (up to ∼100 nm). Cells were first dropped on the coverslip in imaging buffer and given ∼10 min to spread before imaging commenced. Imaging was conducted for 20–30 cells per experiment at a rate of ∼3 min between cells, hence ∼60–90 min typically passed between the first and last cells imaged. Imaging was conducted at room temperature.

For live cell imaging, cells were illuminated by a 561 nm excitation laser at 80% power for 5,000 frames at 85 fps of an EMCCD Ixon$^+$ camera. Fixed cells were further imaged by PALM. For that, they were illuminated using a 405 nm laser for photoactivation of the PAGFP using a changing intensity over the duration of the imaging sequence (typically, using 1–20% power). Illumination with a 488 nm laser at 100% power was used for PAGFP excitation.

**Particle localization.** Single molecule localization was performed using an ImageJ ThunderStorm plugin[64]. The default values were used for the analysis (B-spline wavelet filter—order 3 and scale 2.0, approximate localization by eight-neighbourhood local maximum, Subpixel localization by PSF integrated Gaussian with the weighted least squares fitting method with a 3 pixels fitting radius and 1.6 pixels initial sigma). Particle coordinates and statistical properties were exported and further analysis was conducted using Matlab 2013a (MathWorks). Image drift and vibrations were corrected by mean image displacement values using a custom code in MATLAB.

**SPT.** Linking particle locations into trajectories were done using a modified MATLAB version of refs 34,35. Briefly, the algorithm links particles in a given frame with the particles in the previous one. Particles are linked according to their displacement distance between frames. In cases of ambiguity, a choice has to be made. This is done by minimizing the overall distance travelled by all particles between two consecutive frames: $\sum_{i=1}^{N} \Delta r_i^2$ where $N$ is the number of particles and $\Delta r_i$ is the displacement of each one. The algorithm uses a threshold for the maximum displacement allowed for a particle in a single time step. We used a distance corresponding to three times the expected displacement: $d_{max} = 3\langle|\Delta r|\rangle \approx 3\sqrt{4D\Delta t}$. Assuming a normal diffusion with $D = 0.5$ μm$^2$ s$^{-1}$, at a frame rate of 85 Hz, this corresponds to $d_{max} \approx 459$ nm. We chose for the

algorithm to have 'no memory', that is, if a particle is lost for one frame or more, the linkage is terminated and a new track would begin once it returns. After the linking process is done, only tracks longer than 50 consecutive frames were kept for further analysis. These stringent measures ensure that erroneous linkage of random localizations would be highly suppressed since the probability of persistent linking of randomly occurring points for 50 consecutive frames is negligible at the particle density we work with.

**Immobile trajectories threshold.** As discussed in the 'Results' section, we set a threshold on the ratio of the radius of gyration $R_g$ and the mean step size $\langle|\Delta r|\rangle$. For ideal immobile particles, this ratio is a constant, while for a particle that actually propagates in space the ratio will increase. For an immobile particle, where the apparent motion is due to localization errors, the localizations would be distributed in a 2D Gaussian distribution around the true location, namely, $\mathcal{N}(0, \sigma^2)$. The radius of gyration for a normal distribution with variance $\sigma^2$ is simply $R_g^2 = 2\sigma^2$, which is just the second moment. For the mean step size, we want to compute the expected value: $\langle|\Delta r|\rangle = E\left(\sqrt{(X_1 - X_2)^2 + (Y_1 - Y_2)^2}\right)$, where $X_{1,2}$ and $Y_{1,2}$ are random variables drawn from a normal distribution with mean 0 and variance $\sigma^2$. The addition or subtraction of two such values is a normal distribution with twice the variance of the first: $X_1 - X_2 \sim \mathcal{N}(0, 2\sigma^2)$. Next, the square of a normal distribution is the $\chi^2$ distribution with 1 degree of freedom (DOF): $(X_1 - X_2)^2 \sim 2\sigma^2\chi_1^2$. The sum of $\chi^2$ distributions gives a $\chi^2$ distribution with 2 DOF: $(X_1 - X_2)^2 + (Y_1 - Y_2)^2 \sim 2\sigma^2\chi_2^2$ and the square root then gives the $\chi^2$ with 2 DOF: $\sqrt{(X_1 - X_2)^2 + (Y_1 - Y_2)^2} \sim \sqrt{2}\sigma\chi_2 = \sqrt{2}\sigma x e^{-\frac{x^2}{2}}$. Then, the mean step size, which is the expected value, is $\langle|\Delta r|\rangle = E\left(\sqrt{(X_1 - X_2)^2 + (Y_1 - Y_2)^2}\right) = \int_0^{\infty} \sqrt{2}\sigma x^2 e^{-\frac{x^2}{2}} dx = \sqrt{2}\sigma \cdot \sqrt{2}\Gamma\left(\frac{3}{2}\right) = \sigma\sqrt{\pi}$. Therefore, for a trajectory composed of points taken from a normal distribution with mean 0 and variance $\sigma^2$, the ratio of the radius of gyration and the mean step size is a constant: $R_g/\langle|\Delta r|\rangle = \sqrt{2/\pi}$. Importantly, this ratio is independent of the localization error. Note that the localization error is due to multiple factors[36,37], including factors that depend on the detection system (for example, dark noise, quantization noise and detector pixel size), the sample (for example, background level), as well as particle-related factors (for example, detected fluorescence intensity, its size, velocity and so on).

For fixed cells and for immobile 40 nm gold beads fixed to the coverslip, we find that 95% of the normalized ratio values $\sqrt{\pi/2}R_g/\langle|\Delta r|\rangle$ fall below the value 2.11 and 99% below 3.89. We use the threshold for 5% false positives (that is, 2.11) for the rest of the analyses of the mobile particles. Importantly, this straight-forward method allows the detection of immobile particles, where $R_g$ and $\langle|\Delta r|\rangle$ have large and comparable values. In contrast, using only the diffusion coefficient or $R_g$ for rejecting immobile particles would falsely classify such particles as mobile. Notably, our excluded immobile particles account for 66% of all detected particles.

**Statistical tests and calculations.** The values of α and $K_\alpha$ are calculated by a linear fit to the first 50 time points in any one of the MSDs. Errors for $K_\alpha$ and α are computed by the s.d. of values found between different threshold sets, which met the guidelines of the CER segmentation (see 'Methods' section and Supplementary Fig. 5 for choice of the threshold set for CER segmentation). For the whole ensemble of trajectories, before segmentation, the values are calculated for all time points for $\alpha_e$ and only the linear regime (time points 10–30) for $\alpha_t$. The errors, in this case, are taken from the linear fit while using weights according to the s.e.m. of the values of the MSD curves.

Segmentation is done as described in the main text. Gaps up to three time points of inconsistency in the mobility state are smoothed out. For example, a trajectory with mobility states … 1 1 1 1 1 2 2 1 1 1 1 1 1 … would be smoothed to become all 1s. This is done to minimize noise-related errors in segmentation.

Gaussian fitting for step-size distributions is done based on the least square curve fitting function in MATLAB (Optimization toolbox). The code is adjusted from https://www.mathworks.com/matlabcentral/fileexchange/40613-multiple-curve-fitting-with-common-parameters-using-nlinfit so that boundaries can be introduced. Multiple curves are given simultaneously for fitting with the same set of Gaussian parameters as discussed in the main text.

Segmentation by the vbSPT method was done by a published MATLAB code[25].

Measurement of the fractal dimension on PALM images is done by the box-counting method. MATLAB code was custom written for this.

**Localization error corrections for second and fourth moments.** As shown by Kepten et al.[10], localization errors affect the apparent MSD and may result in an underestimation of α and $K_\alpha$ by raising the MSD at short times. Here we will repeat the calculation done for the MSD by ref. 10 and extend it further to the fourth moment, which we use for the moment ratio test.

Let $x$ and $y$ be the true locations of the particle, and $\varepsilon_x$ and $\varepsilon_y$ be the corresponding localization errors. Then the observed locations are $\hat{x} = x + \varepsilon_x$ and $\hat{y} = y + \varepsilon_y$. We further assume $\langle\varepsilon_x\rangle = \langle\varepsilon_y\rangle = 0$ and $\langle\varepsilon_x^2\rangle = \langle\varepsilon_y^2\rangle = \frac{1}{2}\langle\varepsilon_r^2\rangle$,

where $\vec{r} = \vec{x} + \vec{y}$. The error in the displacement between two locations $x_1$ and $x_2$ due to localization amounts to: $\Delta\hat{x} = \hat{x}_2 - \hat{x}_1 = \Delta x + (\varepsilon_{x,2} - \varepsilon_{x,1})$. Therefore, the observed mean squared displacement is:

$$\langle\Delta\hat{x}^2\rangle = \langle\Delta x^2\rangle + \overbrace{2\langle\Delta x(\varepsilon_{x,2} - \varepsilon_{x,1})\rangle}^{0} + \langle(\varepsilon_{x,2} - \varepsilon_{x,1})^2\rangle$$

$$= \langle\Delta x^2\rangle + \langle\varepsilon_{x,1}^2\rangle + \langle\varepsilon_{x,2}^2\rangle - \overbrace{\langle 2\varepsilon_{x,1}\varepsilon_{x,2}\rangle}^{0} \quad (15)$$

$$= \langle\Delta x^2\rangle + \langle 2\varepsilon_x^2\rangle = \langle\Delta x^2\rangle + \langle\varepsilon_r^2\rangle$$

In two dimensions: $\langle\Delta\hat{r}^2\rangle = \langle\Delta\hat{x}^2\rangle + \langle\Delta\hat{y}^2\rangle = \langle\Delta r^2\rangle + 2\langle\varepsilon_r^2\rangle$. Thus, the correction needed for the MSD is:

$$\langle\Delta r^2\rangle = \langle\Delta\hat{r}^2\rangle - 2\langle\varepsilon_r^2\rangle \quad (16)$$

We now turn to the correction needed for the fourth moment:

$$\langle\Delta\hat{x}^4\rangle = \langle(\hat{x}_2 - \hat{x}_1)^4\rangle = \langle(x_2 - x_1 + (\varepsilon_{x,2} - \varepsilon_{x,1}))^4\rangle = \langle(\Delta x + (\varepsilon_{x,2} - \varepsilon_{x,1}))^4\rangle$$

$$= \langle\Delta x^4\rangle + \overbrace{4\langle\Delta x^3(\varepsilon_{x,2} - \varepsilon_{x,1})\rangle}^{0} + 6\langle\Delta x^2(\varepsilon_{x,2} - \varepsilon_{x,1})^2\rangle$$

$$+ \overbrace{4\langle\Delta x(\varepsilon_{x,2} - \varepsilon_{x,1})^3\rangle}^{0} + \langle(\varepsilon_{x,2} - \varepsilon_{x,1})^4\rangle$$

$$= \langle\Delta x^4\rangle + 6\langle\Delta x^2\rangle\langle(\varepsilon_{x,2} - \varepsilon_{x,1})^2\rangle + \langle(\varepsilon_{x,2} - \varepsilon_{x,1})^4\rangle \quad (17)$$

The last argument on the RHS of equation (17) can be expanded to:

$$\langle(\varepsilon_{x,2} - \varepsilon_{x,1})^4\rangle = \langle\varepsilon_{x,2}^4\rangle - 4\langle\varepsilon_{x,2}^3\rangle\langle\varepsilon_{x,1}\rangle + 6\langle\varepsilon_{x,2}^2\rangle\langle\varepsilon_{x,1}^2\rangle$$

$$- 4\langle\varepsilon_{x,2}\rangle\langle\varepsilon_{x,1}^3\rangle + \langle\varepsilon_{x,1}^4\rangle = 2\langle\varepsilon_x^4\rangle + 6\langle\varepsilon_x^2 2\rangle \quad (18)$$

and the second argument is:

$$6\langle\Delta x^2\rangle\langle(\varepsilon_{x,2} - \varepsilon_{x,1})^2\rangle = 12\langle\Delta x^2\rangle\langle\varepsilon_x^2\rangle \quad (19)$$

Thus, we arrive at:

$$\langle\Delta\hat{x}^4\rangle = \langle\Delta x^4\rangle + 12\langle\Delta x^2\rangle\langle\varepsilon_x^2\rangle + 2\langle\varepsilon_x^4\rangle + 6\langle\varepsilon_x^2\rangle^2 \quad (20)$$

As noted earlier, $\varepsilon_x^2 = \varepsilon_y^2 = \frac{1}{2}\varepsilon_r^2$ and $\varepsilon_x^4$ can be realized by:

$$\langle\varepsilon_r^4\rangle = \langle\varepsilon_x^4\rangle + \langle\varepsilon_y^4\rangle + 2\langle\varepsilon_x^2\rangle\langle\varepsilon_y^2\rangle = 2\langle\varepsilon_x^4\rangle + \frac{1}{2}\langle\varepsilon_r^2\rangle^2 \quad (21)$$

Thus:

$$\langle\varepsilon_x^4\rangle = \frac{1}{2}\langle\varepsilon_r^4\rangle - \frac{1}{4}\langle\varepsilon_r^2\rangle^2 \quad (22)$$

And equation (20) can be written as:

$$\langle\Delta\hat{x}^4\rangle = \langle\Delta x^4\rangle + 6\langle\Delta x^2\rangle\langle\varepsilon_r^2\rangle + \langle\varepsilon_r^4\rangle - \frac{1}{2}\langle\varepsilon_r^2\rangle^2 + \frac{3}{2}\langle\varepsilon_r^2\rangle^2$$

$$= \langle\Delta x^4\rangle + 6\langle\Delta x^2\rangle\langle\varepsilon_r^2\rangle + \langle\varepsilon_r^4\rangle + \langle\varepsilon_r^2\rangle^2 \quad (23)$$

We now note that:

$$\langle\Delta\hat{r}^4\rangle = \langle(\Delta\hat{r}^2)^2\rangle = \langle(\Delta\hat{x}^2 + \Delta\hat{y}^2)^2\rangle$$

$$= \langle\Delta\hat{x}^4\rangle + \langle\Delta\hat{y}^4\rangle + 2\langle\Delta\hat{x}^2\rangle\langle\Delta\hat{y}^2\rangle \quad (24)$$

and so:

$$\langle\Delta\hat{r}^4\rangle = \langle\Delta x^4\rangle + \langle\Delta y^4\rangle + 6(\langle\Delta x^2\rangle + \langle\Delta y^2\rangle)\langle\varepsilon_r^2\rangle + 2\langle\varepsilon_r^4\rangle + 2\langle\varepsilon_r^2\rangle^2$$

$$+ 2(\langle\Delta x^2\rangle + \langle\varepsilon_r^2\rangle)(\langle\Delta y^2\rangle + \langle\varepsilon_r^2\rangle)$$

$$= \langle\Delta x^4\rangle + \langle\Delta y^4\rangle + 6(\langle\Delta x^2\rangle + \langle\Delta y^2\rangle)\langle\varepsilon_r^2\rangle + 2\langle\varepsilon_r^4\rangle + 2\langle\varepsilon_r^2\rangle^2 + 2\langle\Delta x^2\rangle\langle\Delta y^2\rangle$$

$$+ 2(\langle\Delta x^2\rangle + \langle\Delta y^2\rangle)\langle\varepsilon_r^2\rangle + 2\langle\varepsilon_r^2\rangle^2 = \langle\Delta r^4\rangle + 8\langle\Delta r^2\rangle\langle\varepsilon_r^2\rangle + 2\langle\varepsilon_r^4\rangle + 4\langle\varepsilon_r^2\rangle^2 \quad (25)$$

And so finally, the correction for the fourth moment is:

$$\langle\Delta r^4\rangle = \langle\Delta\hat{r}^4\rangle - 8\langle\Delta r^2\rangle\langle\varepsilon_r^2\rangle - 2\langle\varepsilon_r^4\rangle - 4\langle\varepsilon_r^2\rangle^2 \quad (26)$$

The values for $\langle\varepsilon_r^2\rangle$ and $\langle\varepsilon_r^4\rangle$ are taken from the uncertainty estimates given by the ThunderSTORM localization algorithm[64].

**Choosing the threshold set for CER segmentation.** In the case of a three states model, we use a set of four thresholds: $\{R_{th}^{s_1,s_2}, n_{th}^{s_1,s_2}\}, \{R_{th}^{s_2,s_3}, n_{th}^{s_2,s_3}\}$. This means that we are looking for an optimal set in a four-dimensional space. To simplify this process, we fix two thresholds related to the transition between (for example, $s_1,s_2$) and scan a range of relevant values for the other two thresholds (for the same example, between $s_2,s_3$). For each set of thresholds, we get three subpopulations and a matrix of weights $a_i^j$ as defined Sec. IV in the main text. As a reminder, the weight matrix reports on the weight of each of the three Gaussian components fitted for each of the three subpopulations. For clarity, we provide a detailed example of a representative case:

$$a = \begin{bmatrix} 0.95 & 0.05 & 0 \\ 0.03 & 0.92 & 0.05 \\ 0 & 0.07 & 0.93 \end{bmatrix} \quad (27)$$

This matrix reports that the first subpopulation consists of 95% of the low mobility state $s_1$ and 5% of the medium mobility state $s_2$. The second subpopulation consists of a mixture of 3%, 92% and 5% of mobility states $s_1,s_2,s_3$, respectively. The third subpopulation consists of a mixture of 7% and 93% of $s_2$ and $s_3$, respectively. Note that the elements in each row must sum to 1.

On the basis of such a matrix, we compute a score of the extent of mixing between states as follows:

$$score = \sum_{i=j}\left(a_i^j\right)^2 - \sum_{i\neq j}\left(a_i^j\right)^2 \quad (28)$$

In the case of our example, we would have a score of 2.603 out of the possible maximum of 3.0. For a given range of thresholds, we construct a 2D heat map showing the score for each set of thresholds of the variable thresholds, while the two other values are fixed.

Importantly, the total number of segments that are accounted for by each set of thresholds may differ. This is due to the fact that only trajectory segments longer than 50 frames are kept. Thus, long trajectories may be excluded from the analysis if they break down into several short segments that are shorter than 50 frames each. We take this restrictive approach to be able to classify these segments properly in the next stages of the analysis where we perform multiple statistical tests for the identification of the underlying mechanisms. Supplementary Fig. 5a,b shows two representative heat maps. Supplementary Fig. 5a shows a heat map for fixed thresholds $\{R_{th}^{s_1,s_2} = 40\,nm, n_{th}^{s_1,s_2} = 7frames\}$ and variable thresholds $\{R_{th}^{s_2,s_3} \in (115, 150)\,nm, n_{th}^{s_2,s_3} \in (6, 10)frames\}$. Supplementary Figure 5b shows a heat map for variable thresholds $\{R_{th}^{s_1,s_2} \in (35, 60\,nm), n_{th}^{s_1,s_2} \in (5, 9)frames\}$ and fixed thresholds $\{R_{th}^{s_2,s_3} = 125\,nm, n_{th}^{s_2,s_3} = 7frames\}$. For each set, we specify the on-diagonal values $a_{i=j}^j$ and the number of trajectory segments in each subpopulation. Having the scores and the number of segments, we now provide general guidelines for the choice of a preferable threshold set:

1. $R_{th}$ should not be too close to the resolution limit. Here in Supplementary Fig. 5b, we prefer threshold sets where $R_{th} \geq 40$ nm.
2. Sets that result in all three values of $a_{i=j}^j \geq 90$ % are preferable.
3. Sets that result in any of the $a_{i=j}^j = 100$ % should be treated with caution since the fitting algorithm puts a bound on the value which cannot exceed 1.0. Thus, such values may indicate a corrupted fit for which the value would have exceeded 1.0 without the boundary. Hence, we do not consider such sets in our choice.
4. We prefer threshold sets that result in numbers of segments that represent well the distinct subpopulations. For instance, diluting a subpopulation to only its most representative trajectory segments would result in a high fitting score, but with too few segments for further analysis.

On the basis of these guidelines, we now turn to discuss the choice of threshold sets in our data. We observe a tendency of high scores to appear along with a ridge in the heat maps together with a negative diagonal trend. Along this ridge, there are several sets that meet the above guidelines. We tested these sets and found that they overall provide very similar results and underlying mechanisms for the subpopulations. We conclude that our results are insensitive to the exact choice of a threshold set. Supplementary Figure 5c shows $\alpha$ and $K_\alpha$ values for the various threshold sets tested (grey points) together with the threshold sets that met the above guidelines (squares). As can be seen, $\alpha$ and $K_\alpha$ vary in their resultant values. This variability is reduced by a factor of $\sim 3$–4 when using the guidelines. The s.d. of $\alpha$ is typically $\sim 1$–2% of the mean value and $\sim 17\%$ for $K_\alpha$. This high variability for $K_\alpha$ is understandable since it varies over several orders of magnitude. The values we find for $\alpha$ and $K_\alpha$ are summarized in Table 1. Furthermore, we note that, as discussed in the main text, the low and medium subpopulations are in fact part of a continuum of mobility states and the effort for distinct classification is somewhat subjective. To conclude, we recommend testing the robustness of threshold sets in the resultant $\alpha$ and $K_\alpha$ across multiple sets when employing this method.

**Clustering of molecules detected by PALM.** Individual PAGFP-tagged proteins were clustered using a custom code in MATLAB for non-hierarchical clustering. Two molecules were assigned to the same cluster if they were closer than a threshold distance. We chose a distance threshold of 40 nm that accounts for the resolution limit in the localization of two proximal molecules in our study (see mode at 28 nm for the localization uncertainty in Supplementary Fig. 1a). We also tested thresholds between 10 and 60 nm and verified the resemblance of their resultant cluster size distributions, and thus, the insensitivity of this distribution to the chosen threshold.

**Confinement simulations.** Simulations were written in custom MATLAB code. Five hundred particles were simulated for 500 time steps. Confinement was introduced by an un-crossable circular boundary with a radius of 10 units (a.u). Particles were distributed randomly inside the boundary at the initiation of the simulation. For an fBM random walk, the 'wfbm' function was used with an $\alpha$ value of 0.5. The mean step size is $\sim 0.9$ units. Resulting $\alpha$ values are $\alpha_e = 0.56 \pm 0.01$; $\alpha_t = 0.50 \pm 0.01$; $\alpha_{MME} = 0.63 \pm 0.02$. For pure Brownian motion, the mean step size is $\sim 0.4$ and the resulting $\alpha$ values are $\alpha_e = 0.96 \pm 0.08$; $\alpha_t = 0.90 \pm 0.03$; $\alpha_{MME} = 0.98 \pm 0.03$.

These results demonstrate the effects of confinement on random walks with or without an additional subdiffusive mechanism. Importantly, notice the difference between the time averages and the ensemble or MME average values. The choice of simulating fBM rather than RWF is due to the relative simplicity of simulation and demonstration of results.

**Code availability.** All custom codes used in this work are freely available at https://github.com/ShermanLab/SubDiffusion. These codes use MATLAB 2013a (MathWorks).

**Data availability.** The authors declare that all data that supports the findings of this study are available within the article and its Supplementary Information files and from the corresponding author on reasonable request.

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

## Acknowledgements

We thank Yasmine Meroz (Harvard University) and Ralf Metzler (Potsdam University) for fruitful discussions and Roland Schwarzer (Gladstone Institute) and Andreas Herrmann (Humboldt University) for providing gp41 plasmids with fluorescent tags. This research was supported by Grant no. 321993 from the Marie Skłodowska-Curie actions of the European Commission, and Grants no. 1417/13 and no. 1937/13 from the Israeli Science Foundation.

## Author contributions

E.S. supervised research; E.S. and Y.G. designed research; Y.G. performed research; Y.G. and E.S. analysed the data and wrote the paper.

## Additional information

**Competing interests:** The authors declare no competing financial interests.

