## [Peer Review File · Nature Communications]

Reviewers' Comments:

Reviewer #1 (Remarks to the Author)

The manuscript presents some interesting results but the message could be clearer. I am also not convinced the conclusions are well supported.

The manuscript attempts to focus on two different aspects: 1. How to resolve different subdiffusive populations where one of the populations might be from fractional Brownian motion and another from a fractal environment. 2. The spatial heterogeneity with different subdiffusive populations within activated and non-activated T-cells. The manuscript would read better if the introduction defines what is the main problem either part (1) or part (2).

Eq. (4) is defined as an arithmetic mean. However, later it is written this is a geometric mean. Almost all published results work with an arithmetic mean. So the statements about it been a geometric mean is confusing. Ref. 10 shows that using a geometric mean provides a better estimator for alpha. However, using arithmetic mean is not necessarily wrong.

Page 2, model (d). Tracers diffusing in a confined environment will regain normal diffusive behavior at long times if the boundaries are permeable and will saturate to a flat MSD if not.

The Heterogeneous Diffusion Processes are poorly discussed in the introduction. There are many biophysical works that describe them both theoretical and experimental.

What percentage of the trajectories are excluded from the analysis because the particles are found to be immobile?

In section III, the authors conclude the process is ergodic because alpha estimated from time averaged MSD and ensemble averaged MSD are the same. However, K is different so I do not believe ergodicity can be inferred.

Claiming the method of consecutive escape radii is introduced does not seem to be correct. The manuscript should simply state the method is modified because it is actually the same method used in the cited references but with minor modifications.

In section VI. The failure of CTRW model is rejected based on the autocorrelation function. This is simply a confirmation of the failure of the model according to the time averaged MSD.

The "short" time independence in the autocorrelation functions depends on the timescale set by confinement size. This should be quantified.

Describe the meaning of "weakly" confined.

The discussion in section VI claims some similarities with diffusion in a fractal. However the fractal structure in a live cell does not need to be static. Maybe what resembles to be fBM could also be interpreted as diffusion in a dynamic fractal environment. That is a fractal environment with a given dimension but a structure that is reorganized over time.

In general the different methods tested in Section VI seem to contradict each other, and the evidence does not agree with the authors' conclusions. It seems the authors should better test for evidence of subordinated processes. The coexistence of fBM and RWF on the surface of T cells seems to be most likely. However I do not see the evidence provided as confirming this hypothesis.

Minor problems to correct:

1. Equation (1) is a property of Brownian motion, not its definition.
2. After Eq. (4), the ensemble and time averages converge 'to the same value', not simply converge. It is also not required for τ to be large.
3. In page 2, model (b). "The particles may exhibit trapping events with a diverging waiting time distribution." This is written as something trivial. However, trapping events should have exponential waiting times unless the trapping is not Markovian. This is not trivial for biochemical reactions. At least some explanation is needed.

Reviewer #2 (Remarks to the Author)

This is a very interesting work performing thorough analysis of the results of single-particle tracking experiments on the motion of a trans-membrane protein in a cell's plasma membrane combined with photoactivated localization microscopy and uncovering the intra- and intertrajectory heterogeneity in this motion. The work uses a large repertoire of modern statistical approaches and tests, and gives a plausible physical picture of the situation. I would like to see the work published, however, its chaotic style and inconsistent terminology makes a revision necessary.

Let us for example go to Sec. V of the manuscript. The trajectories show considerable intratrajectory variability which is unveiled by segmentation. Each segment of the trajectory is characterized by a relatively well-defined probability density function of the displacements per time step. These probability densities are well fitted by mixtures of Gaussians, and the authors identify three types of such mixtures corresponding to fast, intermediate, and slow motion, which the authors then call the mobility states (e.g. in the caption to Fig. 4). Then the authors turn to the discussion of "subpopulations", and the continuation of the discussion associates these with the size of the diffusing complex. Since the same trajectory shows transition between different mobility states, this size must vary in time, Fig. 4 shows probably that segments with lower mobility occur preferably closer to the center of mass of the cell, while the high-mobility segments are more often seen at the periphery. The continuous change from "mobility states" to "subpopulations" and back, without clearly defining the difference between the notions makes reading very hard. Moreover, the first sentence in the 2nd paragraph on p. 6: "Fig. 4(c,d) shows the spatial distribution of trajectories associated with the 3 mobility states" makes the whole even more obscure: Since the mobility states are associated with the segments of trajectories, what are "trajectories associated with ... mobility states"? The authors should carefully read the whole manuscript and clearly define the notions they use.

The wording used in the discussion at the end of p.4 and around Eq.(7) is misleading because it mixes terminology from two different domains of mathematics. Eq.(7) is not a normal mode decomposition because the Gaussians (normal distributions) are not normal (i.e. orthogonal) to each other. Speaking about the amplitudes and matrix elements is highly misleading, and using the quotation marks ("normal mode") does not save the situation. Eq.(7) represents a (finite) MIXTURE of Gaussians (which should be called the COMPONENTS), and $\sum_i a_i \langle j \rangle$ summing up to unity are the WEIGHTS of the components of the mixture. This is a correct terminology which should be used.

The discussion of the Gaussianity of step size distribution in the right column on p. 4 is extremely obscure, because mixes different notions of steps. In the classical random walk model (e.g. [9]) the step size distribution is quite arbitrary (for example in the Pearson's random walk problem, which was the first random walk ever considered under this name, step length was taken constant). The Gaussian distribution appears after many steps by virtue of CLT. The CLT does not apply to a single step!

The discussion of the fractal distribution of gp41 in the left column of p.7 is not quite clear. It seems that this discussion assumes that this distribution more or less mirrors the fractal dimension of some structure on which the diffusion takes place: particles spread on the structure more or less homogeneously, and thus visualize it. If it is what is meant, than this must be stated clearly.

What do the words "the weak fractal-like structure" mean? If we assume that the fractal dimension of the structure is indeed 1.84 ± 0.07 than it is very close to the one of the percolation cluster (1.89), which is to no extent "weakly fractal". I think that this fact is worth mentioning.

The statement that the result, Eq.(11), for the RWF is exact is wrong. This result is an approximation, as is clearly indicated in Ref. [20] itself, see the sentence before Eq.(7) of the work: "For a random walk on a fractal, approximated by the dynamical scheme of O'Shaughnessy and Procaccia...". The approximation of O'Shaughnessy and Procaccia works well typically (e.g. for percolation) but not universally.

Two minor notes: what does "(Fig.4Fig.2)" at the top of the right column on p.3 refer to? What does the sentence "Note that the localization error is a factor of the system at hand, and of properties of the particle (esp. its size and intensity)." mean? What is the system? The physical or biological system (cell, its membrane), or the data acquisition appliance? What is the intensity of the particle?

Reviewer #3 (Remarks to the Author)

The current study characterizes the mobility of short transmembrane proteins on plasma membranes of living T cells using single particle tracking. The distribution of individual proteins on the membranes was characterised using photo-activated localisation microscopy. The authors detect a highly heterogeneous and ergodic subdiffusive mobility of proteins and set out to uncover the underlying stochastic mechanisms giving rise to the observed motion. The first mechanism they propose is based on the representation of a given trace in terms of multiple Brownian processes with different parameters. The second approach is based on consecutive escape radii (CER) method and involves the statistics of first time passage events. This study introduces novel analysis methods to study the motion of membrane components with sophisticated single particle tracking methods. Such information will ultimately provide novel and indispensable information on the all important cell membrane and the dynamics of its components. This is certainly a topic of such high relevance and timeliness to warrant publication in an outstanding outlet such as Nature Communications. The present work will be an important guide for future experiments, and is poised to strongly influence the growing and highly interdisciplinary field of single particle tracking applications to explore complex biologically relevant systems. I strongly support publication of this work in Nature Communications.

I only have few comments that the authors should address:

(1) As the authors mention, the number of modes in Eq (7) as well as the contribution of each of them to the collection of recorded particle time series is a priori unknown. Specifically, the authors implemented a three-state model with distinct sub-populations of particles with low, medium, and high mobilities. It was found that "particle subpopulations of low and medium mobility consist of protein clusters that diffuse in a viscoelastic and fractal-like medium", while "particles of high mobility undergo weak confinement and are evenly distributed in the cell footprint". Low- and medium-mobility populations are generally claimed to belong to the same mathematical class of subdiffusive models, just with two distinct generalised diffusion coefficients. The authors should clarify for the reader to what extent the conclusions about subpopulations drawn in the manuscript are applicable to other membrane-diffusing particles/proteins and cell types. The authors should also comment in more detail on the physical

reasons why the high-mobility population undergoing a weak confinement still has a subdiffusive MSD scaling, at least in the long-time limit. Does the exponent of the fast subset approach unity at short times?

(2) The distinction of low, medium, and high mobility subpopulations, as extracted from the analysis of autocorrelation functions and presented in Fig. 6, is not very revealing in distinguishing different underlying mathematical models for each sub-population. The three panels in Fig 6 look quite similar in the regime of medium-to-long times. The authors also conclude that non-activated cells become more viscoelastic and less fractal-like, with fractional Brownian motion being the dominant process for such cells. Here, the authors should comment in the revised version in detail on physical and chemical changes taking place on the cell membrane upon this transition and on the effects of these changes onto diffusion mechanisms of the proteins (different binding partners, cell-surface adhesion, cell shape, membrane viscosity for protein diffusion, etc).

(3) How arbitrary is the division of particles into different populations---such as e.g. those presented in Fig 8c---when the boundaries between the subsets are not well recognised over the entire range of diffusion times of recorded trajectories? In the above mentioned figure, for instance, the distinction is clear at short times, whereas in the long time limit individual trajectories overlap substantially. Also, is it generally true that if the overlap of populations is more extensive, such as the one in Fig S3(c) for instance, a model with more than three subpopulations should be used?

(4) The MSD and time averaged MSD for all three subpopulations presented in Fig 5 reveal rather substantial differences in their magnitude, particularly at long diffusion times. Are these differences consistent with the ergodic properties of diffusion models proposed for each subpopulation? The same remark is valid for Fig 8c: a more explicit clarification of ergodicity and equivalence of the two averages should be added here.

(5) The authors might consider moving the rather important Fig 4S(c) to the main text.

Reply to reviewers

We would like to thank the reviewers for their detailed comments on our manuscript. Our point-by point address to the comments is marked below in blue. We feel that these corrections have much improved our manuscript and its readability, towards possible publication.

Please note that in our revised manuscript, the order of multiple supplemental figures have changed. Our answers below always refer to the new figure location, wherever a figure is mentioned. Similarly, since multiple references have been added, the mentioned references below match the updated reference list. Specific textual changes are marked in cyan here, and in the text of the revised manuscript.

Reviewer #1 (Remarks to the Author):

The manuscript presents some interesting results but the message could be clearer. I am also not convinced the conclusions are well supported.

We thank the reviewer for his interest in our results and for his detailed comments. We have now modified the text to make the message of the manuscript clearer. We have also included new simulative results and discussion to support our conclusions.

The manuscript attempts to focus on two different aspects: 1. How to resolve different subdiffusive populations where one of the populations might be from fractional Brownian motion and another from a fractal environment. 2. The spatial heterogeneity with different subdiffusive populations within activated and non-activated T-cells. The manuscript would read better if the introduction defines what is the main problem either part (1) or part (2).

We agree that our introduction may have been confusing regarding the main message of the manuscript. We have now modified the introduction in multiple places to highlight the focus of our work, which is the resolving of mixed sub-diffusion mechanisms in the complex environment of the plasma membrane.

Eq. (4) is defined as an arithmetic mean. However, later it is written this is a geometric mean. Almost all published results work with an arithmetic mean. So the statements about it been a geometric mean is confusing. Ref. 10 shows that using a geometric mean provides a better estimator for alpha. However, using arithmetic mean is not necessarily wrong.

Indeed, our statement describing the averaging in Eq. 4 was wrong. We have now corrected it (to be an arithmetic mean). We have also added a new equation (6) to clearly define the geometric averaging and to clarify the differences between these averaging approaches: "Note should be taken that the mean of the tMSD functions in (4) is an arithmetic mean. In order to provide an accurate estimator for the mean value of the power α , the geometric mean is used instead¹⁰:

$$\begin{aligned} \langle \langle \Delta r^2(\tau) \rangle_{time} \rangle_{ens} &= \left(\prod_{i=1}^N \langle \Delta r^2(\tau) \rangle_{time,i} \right)^{1/N} \\ &= \exp \left(\sum_{i=1}^N \log(\langle \Delta r^2(\tau) \rangle_{time,i}) \right) \end{aligned} \quad (6)$$

The geometric mean is mathematically smaller or equal to the arithmetic mean. Since we use the arithmetic mean for the ensemble average (2) and the geometric mean for the averaged time averages (6), the resultant K_α values for the time averaged measurements are biased to be smaller than the K_α values of the ensemble averages."

We further show, in address to a later comment of this reviewer and of reviewer #3, that the type of averaging has a significant effect on the resultant K_α values when applying the analyses to heterogeneous MSD curves.

Page 2, model (d). Tracers diffusing in a confined environment will regain normal diffusive behavior at long times if the boundaries are permeable and will saturate to a flat MSD if not.

We thank the reviewer for this comment and have now clarified the text accordingly: "(d) Tracers diffusing in a confined environment due to non-permeable physical boundaries demonstrate normal diffusion within the boundaries at short timescales, appear to be subdiffusive in intermediate time scales and will saturate to a flat MSD at long time scales. In the case of permeable boundaries the MSD will regain normal diffusion at long time scales¹⁷."

The Heterogeneous Diffusion Processes are poorly discussed in the introduction. There are many biophysical works that describe them both theoretical and experimental.

Our aim in the introduction has been to briefly mention complex models and contrast them with our approach. To clarify this point, we now modified the text, as follows: "Moreover, particle mobility may be also complicated by static heterogeneity (e.g. particle polydispersity) or spatial inhomogeneity of the medium. **Effective or unified models have been developed to address heterogeneity in SPT results²¹⁻²⁶.**" **We elaborate more on such unified models in the Discussion.**

What percentage of the trajectories are excluded from the analysis because the particles are found to be immobile?

The percentage of excluded trajectories is 66%, which we now mention in the text. This is a very significant fraction, and thus, its rejection confers an undesirable bias to the results.

In section III, the authors conclude the process is ergodic because alpha estimated from time averaged MSD and ensemble averaged MSD are the same. However, K is different so I do not believe ergodicity can be inferred.

Indeed, we have inferred ergodicity from the similarity of the resultant alpha values of the time- and ensemble averaged MSDs. We explain the difference in K_α values as a result of the different (arithmetic vs. geometrical) averaging involved in their calculation. To quantify this significant effect, we have now added simulations of heterogeneous MSDs with (broadly) similar alpha and K_α distributions to those of the data. We show in a new Supplementary Fig. 2 and in its caption that the arithmetic and geometric averaging result in similar alpha values, but differ significantly in their K_α values. In fact, this difference accounts *completely* for the difference in the experimental data, thus further supporting our conclusion of no-ergodicity.

Supplementary Fig. 2 – **Differences between arithmetic and geometric averaging of simulated MSDs** (a,b) Distributions of (a) K_α and (b) α values of simulated mean square displacements (MSDs). K_α values are taken from a log-normal distribution with mean $3.2 \cdot 10^{-13} m^2 s^{-\alpha}$ and std $8.6 \cdot 10^{-13} m^2 s^{-\alpha}$ and α values are taken from a normal distribution with mean 0.7 and std 0.17. These distributions are chosen to broadly resemble the distributions of the experimental data (see Fig. 2(b) in the main text). (c) A log-log plot of 6400 simulated MSDs with respect to time. Shown are MSDs for single trajectories (red lines), The arithmetic mean (blue circles) and the geometric mean (blue triangles) of all simulated MSDs. The K_α values for the arithmetic and geometric means are $3.05 \cdot 10^{-13} m^2 s^{-\alpha}$ and $1.02 \cdot 10^{-13} m^2 s^{-\alpha}$ respectively. The α values for the arithmetic and geometric means are 0.65 and 0.70 respectively.

We further describe this data and our conclusion in the text: " As expected from the differences between the arithmetic and geometric averaging (see eqs. (2) and (6)), we obtain $K_{\alpha,t} = 1.09(1) \cdot 10^{-13} m^2 s^{-\alpha}$ which is smaller than $K_{\alpha,e} = 3.4(1) \cdot 10^{-13} m^2 s^{-\alpha}$. To account for this difference we simulate MSDs with comparable characteristics to the experimental data and find similar differences in K_α values between the arithmetic and geometric means (Supplementary Fig. 2)."

Claiming the method of consecutive escape radii is introduced does not seem to be correct. The manuscript should simply state the method is modified because it is actually the same method used in the cited references but with minor modifications.

Our segmentation approach is a technical implementation (or adaptation) of the existing concept of 'escape radii'. We have now refined the text to better describe the advancement in this approach: " In the second approach for segmentation, we adapt concepts used in "First passage time" or "escape radius" statistics^{19,39,40}, and introduce a modified method we term consecutive escape radii (CER)".

In section VI. The failure of CTRW model is rejected based on the autocorrelation function. This is simply a confirmation of the failure of the model according to the time averaged MSD.

This comment is correct. We have now modified the text to clarify this point: "The low and medium mobility states are similar and both exhibit negative correlations that are insensitive for the time periods δ over a wide range of values. This feature confirms our initial ruling out of CTRW as a possible underlying mechanism and points to either fBM or RWF as the dominating mechanisms of subdiffusion."

The "short" time independence in the autocorrelation functions depends on the timescale set by confinement size. This should be quantified.

Following the reviewer's comment, we now discuss (quantitatively) the relation between the confinement size and the time independence in the autocorrelation function at the initial time periods: "The appearance of growing negative correlations with δ immediately after the initial δ value can be understood from the timescale $\overline{\Delta t}$ needed for a particle to travel the mean step size of the high mobility subpopulation $\overline{\Delta r} \cong 71nm$. This value can be estimated by $\overline{\Delta t} \sim \left(\frac{\overline{\Delta r^2}}{K_\alpha}\right)^{1/\alpha}$, which is $\approx 3ms$. This value is shorter than our time resolution specified above. Thus, we conclude that for small δ values only a small fraction of the particles are affected by the boundary, whereas at growing δ values, an increasing fraction of the particles are affected by it. We refer to this as weak confinement of the high mobility subpopulation. In contrast, strong confinement would result in significant negative auto-correlations at the initial time period δ . Furthermore, we note that our analyses are limited by the trajectory life-times to $\sim 0.6s$. This time frame is likely too short for the trajectory segments to regain apparent Brownian MSDs (Fig. 5) and null velocity auto-correlations at long time periods [Fig. 6 (c)]." **This discussion is also useful in defining the 'weak' confinement, and thus addresses the following comment below.**

Describe the meaning of "weakly" confined.

See definition for 'weak' confinement above, in our address to the previous comment.

The discussion in section VI claims some similarities with diffusion in a fractal. However the fractal structure in a live cell does not need to be static. Maybe what resembles to be fBM could also be interpreted as diffusion in a dynamic fractal environment. That is a fractal environment with a given dimension but a structure that is reorganized over time.

We agree with the reviewer that relatively more complex models may be considered to explain our data. However, there may be many such alternative models with varying degrees of freedom. Specifically, a model describing a dynamic fractal environment would seem to give too many degrees of freedom in comparison to our reductionist approach and models. Also, such a model would likely lead to some ergodicity breaking (such as in ref [22]), which we do not detect in our data.

In general the different methods tested in Section VI seem to contradict each other, and the evidence does not agree with the authors' conclusions. It seems the authors should better test for evidence of subordinated processes. The coexistence of fBM and RWF on the surface of T cells seems to be most likely. However I do not see the evidence provided as confirming this hypothesis.

Indeed, it is inherently difficult to resolve fBM from RWF using the statistical tools that we have employed in this study. However, to our knowledge (based on an extensive literature survey and consultation with experts in the field), there seem to be no additional statistics of relevance. With the statistical tools at hand, our results (that now include added simulations in the new Supplementary Fig. 2) strongly support the prevalence of fBM and RWF as underlying mechanisms that affect the low and medium mobility subpopulations. More complicated mechanisms could be considered, including subordinated mechanisms. We examine and discuss such mechanisms in the Discussion section of the manuscript. However, the discussed models do not agree with our data. In future work, we will continue to explore the benefit of alternative statistical tools and mixed mechanisms, yet these efforts are essentially 'open-ended' and exceed the scope of the current work.

Minor problems to correct:

1. Equation (1) is a property of Brownian motion, not its definition.

We have modified the text to address this point.

2. After Eq. (4), the ensemble and time averages converge 'to the same value', not simply converge. It is also not required for tau to be large.

Text Corrected.

3. In page 2, model (b). "The particles may exhibit trapping events with a diverging waiting time distribution." This is written as something trivial. However, trapping events should have exponential waiting times unless the trapping is not Markovian. This is not trivial for biochemical reactions. At least some explanation is needed.

Corrected text to: "The particles may exhibit trapping events with a heavy-tailed waiting time distribution."

Reviewer #2 (Remarks to the Author):

This is a very interesting work performing thorough analysis of the results of single-particle tracking experiments on the motion of a trans-membrane protein in a cell's plasma membrane combined with photoactivated localization microscopy and uncovering the intra- and intertrajectory heterogeneity in this motion. The work uses a large repertoire of modern statistical approaches and tests, and gives a plausible physical picture of the situation. I would like to see the work published, however, its chaotic style and inconsistent terminology makes a revision necessary.

We much appreciate the reviewer's encouraging comments regarding the scope and depth of our analyses, his interest in the work and his support in publication of the work. We have now modified the text and multiple terms within to make it clearer.

Let us for example go to Sec. V of the manuscript. The trajectories show considerable intrajourney variability which is unveiled by segmentation. Each segment of the trajectory is characterized by a relatively well-defined probability density function of the displacements per time step. These probability densities are well fitted by mixtures of Gaussians, and the authors identify three types of such mixtures corresponding to fast, intermediate, and slow motion, which the authors then call the mobility states (e.g. in the caption to Fig. 4). Then the authors turn to the discussion of "subpopulations", and the continuation of the discussion associates these with the size of the diffusing complex. Since the same trajectory shows transition between different mobility states, this size must vary in time, Fig. 4 shows probably that segments with lower mobility occur preferably closer to the center of mass of the cell, while the high-mobility segments are more often seen at the periphery. The continuous change from "mobility states" to "subpopulations" and back, without clearly defining the difference between the notions makes reading very hard. Moreover, the first sentence in the 2nd paragraph on p. 6: "Fig. 4(c,d) shows the spatial distribution of trajectories associated with the 3 mobility states" makes the whole even more obscure: Since the mobility states are associated with the segments of trajectories, what are "trajectories associated with ... mobility states"? The authors should carefully read the whole manuscript and clearly define the notions they use.

We agree with the reviewer that our terminology has not been consistent throughout the text. Thus, we added a paragraph that aims to clarify this terminology: "At this point, we would like to clarify the relation between different terms used in the text that are invoked through our segmentation and classification processes. Through these processes, segment subpopulations are uniquely assigned to specific mobility states. Thus we will now interchangeably use the terms 'mobility states', 'mobility subpopulations' or simply 'subpopulations' to describe the segments of common mobility characteristics (being either 'low', 'medium' or 'high'). Moreover, note should be taken that through the segmentation process, a trajectory of a single particle may be divided into several segments that match different mobility states and may thus be assigned to several segment subpopulations."

Furthermore, we clarify the relation of these terms to the physical characteristics of detected tracer particles (in Fig. 4): "Fig. 4(a) shows the normalized distributions of fluorescence intensity of the 3 subpopulations. This is not a direct characterization of single particles, but rather of their segmented trajectories that might belong to different subpopulations. However, for physically stable particles (i.e. they do not disintegrate or assemble over the typical trajectory duration [Supplementary Fig. 1(c)]), we can infer on average the particle fluorescence intensity distributions from this analysis. Also, we note that although the fluorescence intensity is not a precise measure of the number of proteins in individual particles, it can give a rough estimate of this number." **We then conclude that** "As can be seen in Fig. 4(a), segments belonging to brighter particles are of a lower mobility state on average." **And that** "Fig. 4(c,d) shows the spatial distribution of trajectory segments associated with the 3 mobility states. We find that the higher the mobility of a segment, the farther its corresponding particle can distribute from the center of the cell footprint on average."

The wording used in the discussion at the end of p.4 and around Eq.(7) is misleading because it mixes terminology from two different domains of mathematics. Eq.(7) is not a normal mode decomposition because the Gaussians (normal distributions) are not normal (i.e. orthogonal) to each other. Speaking about the amplitudes and matrix elements is highly misleading, and using the quotation marks ("normal mode") does not save the situation. Eq.(7) represents a (finite) MIXTURE of Gaussians (which should be called the COMPONENTS), and $\langle a_j \rangle$ summing up to unity are the WEIGHTS of the components of the mixture. This is a correct terminology which should be used.

We thank the reviewer for the suggested modifications to our terminology. We have adopted the suggestions and incorporated them, concerning Eq. 7 and throughout the main and supplemental texts, where relevant.

The discussion of the Gaussianity of step size distribution in the right column on p. 4 is extremely obscure, because mixes different notions of steps. In the classical random walk model (e.g. [9]) the step size distribution is quite arbitrary (for example in the Pearson's random walk problem, which was the first random walk ever considered under this name, step length was taken constant). The Gaussian distribution appears after many steps by virtue of CLT. The CLT does not apply to a single step!

We thank the reviewer for pointing out this obscurity and corrected the text as suggested

"Random walks originating from a distinct process effectively yield a Gaussian distribution of step-sizes. The Gaussian distribution appears after many steps of the random walk by virtue of the central limit theorem (CLT)⁴¹, regardless of the original distribution of step-sizes of the underlying process."

The discussion of the fractal distribution of gp41 in the left column of p.7 is not quite clear. It seems that this discussion assumes that this distribution more or less mirrors the fractal dimension of some structure on which the diffusion takes place: particles spread on the structure more or less homogeneously, and thus visualize it. If it is what is meant, than this must be stated clearly.

We have now clarified this meaning in the text, by adding the following sentence: "We assume the particles are homogeneously spread over the structure imaged by PALM, and thus is effectively captured by it."

What do the words "the weak fractal-like structure" mean? If we assume that the fractal dimension of the structure is indeed 1.84 ± 0.07 than it is very close to the one of the percolation cluster (1.89), which is to no extent "weakly fractal". I think that this fact is worth mentioning.

We agree with the reviewer that our fractal is significantly different than 2D, and thus, have excluded the word 'weak' in reference to this fractal. We also agree that the similarity of the measured fractal to the percolation cluster is very interesting. Since we do not explore this fact further in this study, we would prefer to leave this insight for a more thorough study in the future without mentioning a suggestive mechanism.

The statement that the result, Eq.(11), for the RWF is exact is wrong. This result is an approximation, as is clearly indicated in Ref. [20] itself, see the sentence before Eq.(7) of the work: "For a random walk on a fractal, approximated by the dynamical scheme of O'Shaughnessy and Procaccia...". The approximation of O'Shaughnessy and Procaccia works well typically (e.g. for percolation) but not universally.

Indeed, we have mistakenly used the word "exact" here. We now refer to it in the text as an approximation.

Two minor notes: what does "(Fig.4Fig.2)" at the top of the right column on p.3 refer to?

We corrected the text, which now specifies Fig. 2(a).

What does the sentence "Note that the localization error is a factor of the system at hand, and of properties of the particle (esp. its size and intensity)." mean? What is the system? The physical or biological system (cell, its membrane), or the data acquisition appliance? What is the intensity of the particle?

To clarify our meaning, we have now added the following sentence to the supplemental text (section on Methods). "Note that the localization error is due to multiple factors^{4,5}, including factors that depend on the detection system (e.g. dark noise, quantization noise and detector pixel size), the sample (e.g. background level), as well as particle related factors (e.g. detected fluorescence intensity, its size, velocity, etc.)."

Reviewer #3 (Remarks to the Author):

The current study characterizes the mobility of short transmembrane proteins on plasma membranes of living T cells using single particle tracking. The distribution of individual proteins on the membranes was characterised using photo-activated localisation microscopy. The authors detect a highly heterogeneous and ergodic subdiffusive mobility of proteins and set out to uncover the underlying stochastic mechanisms giving rise to the observed motion. The first mechanism they propose is based on the representation of a given trace in terms of multiple Brownian processes with different parameters. The second approach is based on consecutive escape radii (CER) method and involves the statistics of first time passage events. This study introduces novel analysis methods to study the motion of membrane components with sophisticated single particle tracking methods. Such information will ultimately provide novel and indispensable information on the all important cell membrane and the dynamics of its components. This is certainly a topic of such high relevance and timeliness to warrant publication in an outstanding outlet such as Nature Communications. The present work will be an important guide for future experiments, and is poised to strongly influence the growing and highly interdisciplinary field of single particle tracking applications to explore complex biologically relevant systems. I strongly support publication of this work in Nature Communications.

We thank the reviewer for his interest in our study and for his strong support in its publication.

I only have few comments that the authors should address:

(1) As the authors mention, the number of modes in Eq (7) as well as the contribution of each of them to the collection of recorded particle time series is a priori unknown. Specifically, the authors implemented a three-state model with distinct sub-populations of particles with low, medium, and high mobilities. It was found that "particle subpopulations of low and medium mobility consist of protein clusters that diffuse in a viscoelastic and fractal-like medium", while "particles of high mobility undergo weak confinement and are evenly distributed in the cell footprint". Low- and medium-mobility populations are generally claimed to belong to the same mathematical class of subdiffusive models, just with two distinct generalised diffusion coefficients. The authors should clarify for the reader to what extent the conclusions about subpopulations drawn in the manuscript are applicable to other membrane-diffusing particles/proteins and cell types.

We thank the reviewer for raising this important point of generality of our findings. To match the flow of the discussion, we start the answer below by addressing the issue of changes in the plasma-membrane upon cell activation (appearing in comment #2 below), and then relate it to our findings.

"Indeed, cell activation has been shown to cause global changes in the morphology of the PM and the cytoskeleton^{2,3}. These changes include coalescence of signaling proteins into micro-clusters, polymerization of cytoskeletal proteins, including cortical actin and tubulin, tighter adhesion with the antigen presenting cell and spreading. These changes might alter the diffusive motion of proteins at the PM. For instance, signaling proteins might get trapped in clusters³⁷ or in actin corrals⁵⁶, experience diffusion barriers⁵⁸, while the PM might change its three dimensional patterning⁵⁹. Here, we use a relatively inert transmembrane protein, namely the transmembrane domain of gp41 with a mutation that suppresses its interactions with specific lipids³² and cellular proteins at the PM. However, specific interactions of this protein with the PM environment cannot be completely ruled out. Naturally, other transmembrane proteins

in the cell may undergo significant and specific interactions to facilitate their action, and thus may exhibit different underlying subdiffusive mechanisms than found here (e.g. non-ergodic processes)³⁸.”

In the context of our address to this comment, we would like to stress the conclusion in the text: “More generally, our approach and results demonstrate how one can methodologically resolve mixed mechanisms of subdiffusion that act simultaneously on a polydispersed sample within the complex medium of the cell PM. This approach can also serve as a framework for resolving mixed mechanisms in a wide range of random walks of arbitrary nature.”

The authors should also comment in more detail on the physical reasons why the high-mobility population undergoing a weak confinement still has a subdiffusive MSD scaling, at least in the long-time limit. Does the exponent of the fast subset approach unity at short times?

These are interesting points, which have also been raised by reviewer #1. To address these points, we first describe the expected MSDs from confinement in the Introduction: “(d) Tracers diffusing in a confined environment due to non-permeable physical boundaries demonstrate normal diffusion within the boundaries at short timescales, appear to be subdiffusive in intermediate time scales and will saturate to a flat MSD at long time scales. In the case of permeable boundaries the MSD will regain normal diffusion at long time scales¹⁷.”

We then discuss how the confinement affects the high mobility subpopulation in our data, the MSDs of this subpopulation shown in Fig. 5, and the confinement effect on the corresponding VAC results in Fig. 6(c).

“The appearance of growing negative correlations with δ immediately after the initial δ value can be understood from the timescale $\overline{\Delta t}$ needed for a particle to travel the mean step size of the high mobility subpopulation $\overline{\Delta r} \cong 71nm$. This value can be estimated by $\overline{\Delta t} \sim \left(\frac{\overline{\Delta r}^2}{K\alpha}\right)^{1/\alpha}$, which is $\approx 3ms$. This value is shorter than our time resolution specified above. Thus, we conclude that for small δ values only a small fraction of the particles are affected by the boundary, whereas at growing δ values, an increasing fraction of the particles are affected by it. We refer to this as weak confinement of the high mobility subpopulation. In contrast, strong confinement would result in significant negative auto-correlations at the initial time period δ . Furthermore, we note that our analyses are limited by the trajectory life-times to $\sim 0.6s$. This time frame is likely too short for the trajectory segments to regain apparent Brownian MSDs (Fig. 5) and null velocity auto-correlations at long time periods [Fig. 6 (c)].”

Looking closer at our data following the reviewers comment, we observe that the MSD of the high mobility subpopulation still has a subdiffusive exponent at the shortest times of Fig. 5. This measurement is noisy due to the limited data. The subdiffusive exponent might be related to the fact that a substantial number of particles is already affected by confinement, which we discuss in the text in response to a previous comment by reviewer #1.

(2)The distinction of low, medium, and high mobility subpopulations, as extracted from the analysis of autocorrelation functions and presented in Fig. 6, is not very revealing in distinguishing different underlying mathematical models for each sub-population. The three panels in Fig 6 look quite similar in the regime of medium-to-long times.

Indeed, the velocity auto-correlation functions are most revealing in their short time periods and their overall behavior for the negative VAC values. While the low and medium mobility subpopulations (panels a,b) have a flat negative VAC for all time periods, the high mobility subpopulations (panel c) has an increasing negative VAC with growing time periods. These features correspond to the different underlying mechanisms following Weber et al. (ref. [29]).

The authors also conclude that non-activated cells become more viscoelastic and less fractal-like, with fractional Brownian motion being the dominant process for such cells. Here, the authors should comment in the revised version in detail on physical and chemical changes taking place on the cell membrane upon this transition and on the effects of these changes onto diffusion mechanisms of the proteins (different binding partners, cell-surface adhesion, cell shape, membrane viscosity for protein diffusion, etc).

See our response to comment (1) of this reviewer.

(3) How arbitrary is the division of particles into different populations---such as e.g. those presented in Fig 8c---when the boundaries between the subsets are not well recognised over the entire range of diffusion times of recorded trajectories? In the above mentioned figure, for instance, the distinction is clear at short times, whereas in the long time limit individual trajectories overlap substantially. Also, is it generally true that if the overlap of populations is more extensive, such as the one in Fig S3(c) for instance, a model with more than three subpopulations should be used?

A main objective of our study is the process of segmentation and classification of trajectories with minimal arbitrariness in the number of models and in the thresholds that separate the resulting subpopulations. For this we employed 2 different approaches for segmentation and classification, namely the CER and vbSPT. We start with a minimal number of possible mobility states to account for the data and then increase the number of states. In each step we test the results for redundancy of states using multiple statistical characteristics of the segment subpopulations (including the MSDs, VACs, moment ratios and growing sphere analysis; e.g. Fig. 8). Due to statistical variations, some overlap of the MSDs is expected through this process. From the 3 states model and onward, the high mobility subpopulation separates well from the other lower mobility subpopulations. The remaining subpopulations of lower mobilities differ predominantly in their K_α values, but otherwise share multiple statistical characteristics. This strongly suggests that they belong to a common subpopulation of particles, as we conclude in the text. This explains the significant overlap of the trajectories in the corresponding MSD plots of the lower mobility states [Figs 8(c), S3(c), S4(c)].

(4) The MSD and time averaged MSD for all three subpopulations presented in Fig 5 reveal rather substantial differences in their magnitude, particularly at long diffusion times. Are these differences consistent with the ergodic properties of diffusion models proposed for each subpopulation? The same remark is valid for Fig 8c: a more explicit clarification of ergodicity and equivalence of the two averages should be added here.

This point has been raised by reviewer #1 as well and we present an identical response here:

Indeed, we have inferred ergodicity from the similarity of the resultant alpha values of the time- and ensemble averaged MSDs. We explain the difference in K_α values as a result of the different (arithmetic vs. geometrical) averaging involved in their calculation. To quantify this significant effect, we have now added simulations of heterogeneous MSDs with (broadly) similar alpha and K_α distributions to those of the data. We show in a new Supplementary Fig. 2 and in its caption that the arithmetic and geometric averaging result in similar alpha values, but differ significantly in their K_α values. In fact, this difference accounts *completely* for the difference in the experimental data, thus further supporting our conclusion of no-ergodicity.

Supplementary Fig. 2 – Differences between arithmetic and geometric averaging of simulated MSDs
 (a,b) Distributions of (a) K_α and (b) α values of simulated mean square displacements (MSDs). K_α values are taken from a log-normal distribution with mean $3.2 \cdot 10^{-13} m^2 s^{-\alpha}$ and std of $8.6 \cdot 10^{-13} m^2 s^{-\alpha}$ and α values are taken from a normal distribution with mean 0.7 and std of 0.17. These distributions are chosen to broadly resemble the distributions of the experimental data (see Fig. 2(b) in the main text). (c) A log-log plot of 6400 simulated MSDs with respect to time. Shown are MSDs for single trajectories (red lines), The arithmetic mean (blue circles) and the geometric mean (blue triangles) of all simulated MSDs. The K_α values for the arithmetic and geometric means are $3.05 \cdot 10^{-13} m^2 s^{-\alpha}$ and $1.02 \cdot 10^{-13} m^2 s^{-\alpha}$ respectively. The α values for the arithmetic and geometric means are 0.65 and 0.70 respectively.

We describe this data and our conclusion in the text: " As expected from the differences between the arithmetic and geometric averaging (see eqs. (2) and (6)), we obtain $K_{\alpha,t} = 1.09(1) \cdot 10^{-13} m^2 s^{-\alpha}$ which is smaller than $K_{\alpha,e} = 3.4(1) \cdot 10^{-13} m^2 s^{-\alpha}$. To account for this difference we simulate MSDs with comparable characteristics to the experimental data and find similar differences in K_α values between the arithmetic and geometric means (Supplementary Fig. 2)."

(5) The authors might consider moving the rather important Fig 4S(c) to the main text.

We appreciate the recognition of the reviewer of the importance of Fig. S4(c). However, we note that this figure is very specific to the CER classification process and is indicative of the effective errors one might expect from this procedure in the average α and K_α values of the subpopulations. We think that this technical issue would be too lengthy to explain in the main text and thus would prefer to leave it as a supplement.

Reviewers' Comments:

Reviewer #1:

Remarks to the Author:

I am very satisfied with the revised version of the manuscript and I support its publication in Nature Communications. However I still have an issue with the conclusions that the trajectories are ergodic. I greatly appreciate the new provided simulations showing that the differences in generalized diffusion coefficient obtained from time and ensemble averages using geometric and arithmetic means respectively, are not sufficient to show ergodicity breaking. Nevertheless, I believe the data provided neither prove or disprove ergodicity.

I do not think the manuscript needs to show the trajectories are ergodic. The authors can simply claim that the data is consistent with ergodic mechanisms as suggested by numerical simulations, but not that they show the trajectories are ergodic. If the authors wish to prove ergodicity I suggest one of two different approaches: evaluation of the ergodicity breaking parameter (see He, Burov, Metzler and Barkai, Phys. Rev. Lett. 2008) or evaluation of the dynamical functional (see Lanoiselée and Grebenkov, Phys. Rev. E 2016).

Additional minor suggestions that should be considered:

- In the introduction it is mentioned that "particles in crowded or labyrinth-like environments demonstrate diffusion in a fractal-like space". It should say "obstructed" instead of "crowded". Crowded environments can be modeled by fBM (Weiss, Phys. Rev. E 2013).
- A recent article has shown that protein motion in the plasma membrane is described by diffusion in a fractal-like space due the interactions with the actin cytoskeleton (Sadegh, Higgins, Mannion, Tamkun, and Krapf, Phys. Rev. X 2017) providing evidence for a mechanism that explains diffusion in a fractal on the surface of the cell.

Reviewer #2:

Remarks to the Author:

Already reviewing the first version of the manuscript I stressed that it reports on a very interesting and potentially important research, and stated that I would be happy to see the work published. All my remarks were on the style and inconsistent use of terminology. The authors have taken all my remarks into account, and amended the manuscript accordingly. Moreover, they have adequately responded to the criticisms of other referees. I strongly recommend the manuscript for publication now.

Reviewer #3:

Remarks to the Author:

The authors revised their manuscript considerably, following the comments of the Referees. I find the rebuttal fully convincing and have no further objections against publication of this work in Nature Comm.

The topic covered in this works is of such high relevance and timeliness, with high expected impact into this very active field. In my opinion it certainly warrants publication on the outstanding platform offered by Nature Communications.

Reply to reviewers

We would like to thank again the reviewers for their comments, which have considerably improved our manuscript. We also much appreciate their kind and on-going support in the publication of our study. Our point-by point address to the comments is marked below in blue.

Reviewer #1 (Remarks to the Author):

I am very satisfied with the revised version of the manuscript and I support its publication in Nature Communications. However I still have an issue with the conclusions that the trajectories are ergodic. I greatly appreciate the new provided simulations showing that the differences in generalized diffusion coefficient obtained from time and ensemble averages using geometric and arithmetic means respectively, are not sufficient to show ergodicity breaking. Nevertheless, I believe the data provided neither prove or disprove ergodicity.

I do not think the manuscript needs to show the trajectories are ergodic. The authors can simply claim that the data is consistent with ergodic mechanisms as suggested by numerical simulations, but not that they show the trajectories are ergodic. If the authors wish to prove ergodicity I suggest one of two different approaches: evaluation of the ergodicity breaking parameter (see He, Burov, Metzler and Barkai, Phys. Rev. Lett. 2008) or evaluation of the dynamical functional (see Lanoiselée and Grebenkov, Phys. Rev. E 2016).

We thank the reviewer for his support in publication. We agree with the reviewer that our data does not conclusively show the protein trajectories are ergodic. Taking the reviewer's advice, we have modified the text accordingly, and maintain that the data and supportive simulations are consistent with underlying ergodic mechanisms.

Additional minor suggestions that should be considered:

- In the introduction it is mentioned that "particles in crowded or labyrinth-like environments demonstrate diffusion in a fractal-like space". It should say "obstructed" instead of "crowded". Crowded environments can be modeled by fBM (Weiss, Phys. Rev. E 2013).

As suggested, we have replaced the word 'crowded' with 'obstructed'.

- A recent article has shown that protein motion in the plasma membrane is described by diffusion in a fractal-like space due the interactions with the actin cytoskeleton (Sadegh, Higgins, Mannion, Tamkun, and Krapf, Phys. Rev. X 2017) providing evidence for a mechanism that explains diffusion in a fractal on the surface of the cell.

We have added a citation to this recent work by Sadesh et al in our discussion.

Reviewer #2 (Remarks to the Author):

Already reviewing the first version of the manuscript I stressed that it reports on a very interesting and potentially important research, and stated that I would be happy to see the work published. All my remarks were on the style and inconsistent use of terminology. The authors have taken all my remarks into account, and amended the manuscript accordingly. Moreover, they have adequately responded to the criticisms of other referees. I strongly recommend the manuscript for publication now.

We much appreciate the kind words, detailed comments in the last revision cycle, and on-going support in publication by reviewers #2 and 3.

Reviewer #3 (Remarks to the Author):

The authors revised their manuscript considerably, following the comments of the Referees. I find the rebuttal fully convincing and have no further objections against publication of this work in Nature Comm.

The topic covered in this works is of such high relevance and timeliness, with high expected impact into this very active field. In my opinion it certainly warrants publication on the outstanding platform offered by Nature Communications.

We much appreciate the kind words, detailed comments in the last revision cycle, and on-going support in publication by reviewers #2 and 3.